# Score-Based Diffusion Policy
# Compatible with Reinforcement Learning via Optimal Transport

**Mingyang Sun** [1 2 3]  **Pengxiang Ding** [1 2]  **Weinan Zhang** [4 3]  **Donglin Wang** [2]

## Abstract

Diffusion policies have shown promise in learning complex behaviors from demonstrations, particularly for tasks requiring precise control and long-term planning. However, they face challenges in robustness when encountering distribution shifts. This paper explores improving diffusion-based imitation learning models through online interactions with the environment. We propose OTPR (Optimal Transport-guided score-based diffusion Policy for Reinforcement learning fine-tuning), a novel method that integrates diffusion policies with RL using optimal transport theory. OTPR leverages the Q-function as a transport cost and views the policy as an optimal transport map, enabling efficient and stable fine-tuning. Moreover, we introduce masked optimal transport to guide state-action matching using expert keypoints and a compatibility-based resampling strategy to enhance training stability. Experiments on three simulation tasks demonstrate OTPR's superior performance and robustness compared to existing methods, especially in complex and sparse-reward environments. In sum, OTPR provides an effective framework for combining IL and RL, achieving versatile and reliable policy learning.

## 1. Introduction

Robotic manipulation is an intricate endeavor, where the delicate interplay of long-term planning and instantaneous control poses a captivating challenge - the quest to develop policies that can seamlessly navigate this balance lies at the forefront of modern robotics (Heo et al., 2023; Mu et al., 2021; Chen et al., 2023). Tasks demand not only the abil-ity to execute complex sequences, but also the adaptability to handle uncertainties and disturbances. Imitation Learning (IL) has emerged as a popular data-driven approach for training robots by imitating demonstration data, with advancements of Behavior Cloning (BC) like diffusion models (Chi et al., 2023; Ajay et al.) and action chunking (Zhao et al., 2023) enhancing its ability to learn complex, long-horizon behaviors. Notably, Diffusion Policy (DP) (Chi et al., 2023) has shown promise due to the capacity to handle multi-modal action distributions, excel in high-dimensional spaces, and achieve stable training through techniques like denoising and score matching. However, these advancements still fail to address the fundamental flaws of BC, which remains highly susceptible to distributional shifts, where the policy encounters states outside its training data, leading to compounding errors (Ross & Bagnell, 2010).

Reinforcement Learning (RL) offers a powerful framework for autonomous learning through trial-and-error interactions guided by reward signals, making it particularly effective in training reactive controllers that adapt to noise, disturbances, and unforeseen states (Kober et al., 2013). RL learns corrective behaviors directly from experience, enabling policies to recover from errors and handle states beyond the training distribution. Its ability to optimize over long time horizons can also refine action sequences, enhancing robustness and precision. Unlike IL, which benefits from leveraging demonstration data to jump start learning, RL enhances generalization by exploring diverse scenarios and adapting dynamically to environmental changes. However, RL also faces significant challenges, including the need for carefully designed reward functions and vast interaction data, which is costly to collect, particularly in real-world settings (Park et al., 2024).

These strengths and weaknesses suggest that an integrated approach, combining RL's adaptability with DP's demonstration-driven learning, holds promise for achieving reliable, scalable, and versatile robotic manipulation. The most common approach is to pretrain a imitation policy with human data and then finetune it with RL (Black et al., 2024). Some methods apply additional regularization (Rajeswaran et al., 2018) or seperated policy network (Ankile et al., 2024) to ensure that the knowledge from demonstrations

[1]Zhejiang University, Hangzhou, China [2]Westlake University, Hangzhou, China [3]Shanghai Innovation Institute, Shanghai, China [4]Shanghai Jiao Tong University, Shanghai, China. Correspondence to: Donglin Wang <wangdonglin@westlake.edu.cn>.

*Proceedings of the 42nd International Conference on Machine Learning*, Vancouver, Canada. PMLR 267, 2025. Copyright 2025 by the author(s).

does not get washed out quickly by the randomly initialized critics, which may suffer from hyper-parameter tuning issue. Additionally, the structure of diffusion models (iterative refinement) inherently complicate the application of standard RL algorithms, often leading to low-efficiency, instability or requiring significant architectural modifications (Ren et al., 2024; Mark et al., 2024; Psenka et al., 2024).

In this paper, we integrate insights from the optimal transport theory (Gu et al., 2023; Montesuma et al., 2024) to refine the diffusion policy optimization process, leveraging knowledge gained from expert trajectories to improve learning efficiency and policy performance in subsequent reinforcement learning tasks. By utilizing the Q-function as a transport cost and viewing the policy as an optimal transport map, we establish an equivalent relationship between the optimal transport map and the optimal policy, which opens avenues for applying recent advantages of RL to diffusion policy directly. Our key contributions are as follows:

- We proposed an **O**ptimal **T**ransport guided score-based diffusion **P**olicy for **R**einforcement Learning fine-tuning (**OTPR**), which is the first work to systematically combine optimal transport theory with diffusion policies for reinforcement learning fine-tuning. OTPR's core lies in solving the $L_2$-regularized OT dual problem, thereby deriving a compatibility function that establishes a soft coupling relationship between states and actions, effectively integrating imitation learning and reinforcement learning.

- To enhance efficiency and accuracy, we introduce the **Masked Optimal Transport** to leverage the paired state-action from expert data as keypoints to guide the matching of the other state-action points from replay buffer, which seamlessly integrates imitation learning and reinforcement learning objective.

- To address the sub-optimal performance of the conditional score-based model when trained with standard algorithms on mini-batch data, we introduce a **Compatibility-based Resampling** strategy to selects action with high compatibility scores to guide the training process, thereby enhancing performance.

We conduct extensive experiments on 3 simulation tasks spanning various difficulty levels. The results demonstrate that OTPR consistently matches or outperforms existing state-of-the-art methods in all tasks, with particularly notable improvements in more challenging scenarios.

## 2. Related Work

**Diffusion based policies.** Diffusion-based policies have shown recent success in robotics and decision-making applications. In a pioneering work, "Diffuser" (Janner et al.,

2022), a planning algorithm with diffusion models for offline reinforcement learning. This framework is extended to other tasks in the context of offline reinforcement learning (Wang et al., 2022), where the training dataset includes reward values. Most typically, diffusion based policies are trained from human demonstrations through a supervised objective, and enjoy both high training stability and strong performance in modeling complex and multi-modal trajectory distributions. The application of DDPM (Ho et al., 2020) and DDIM (Song et al., 2020a) on visuomotor policy learning for physical robots (Chi et al., 2023; Gong et al., 2024; Jia et al., 2024) outperforms counterparts like Behavioral Cloning. While these techniques effectively learn from multi-modal data, they often create models that are non-trivial to fine-tune using RL. Even if they were compatible with RL, the fine-tuning process can be computationally prohibitive due to the large number of parameters in modern policy models.

**Training diffusion models with reinforcement learning.** As demonstration data are often limited, there have been many approaches proposed to improve the performance of diffusion-based policies. One straightforward approach (Black et al., 2024; Fan et al., 2024) involves framing diffusion denoising as a Markov Decision Process (MDP), which facilitates preference-aligned generation with policy gradient reinforcement learning. However, this approach often suffers from instability, limiting its practical applicability. (Ren et al., 2024) introduced policy gradient loss on a two-layer MDP for direct diffusion policy fine-tuning, which mitigates this instability, but the method is architecture-specific and does not introduce closed-loop control. Alternative approaches to integrating diffusion architectures with reinforcement learning (RL) include leveraging Q-function-based importance sampling (Hansen-Estruch et al., 2023), employing advantage-weighted regression (Goo & Niekum, 2022), or reformulating the objective as a supervised learning problem with return conditioning (Chen et al., 2021; Janner et al., 2022; Ajay et al.). Additionally, researchers have explored enhancing the denoising training objective by incorporating Q-function maximization (Wang et al., 2022) and iteratively refining the dataset using Q-functions (Yang et al., 2023). Another promising direction involves augmenting a frozen, chunked diffusion policy model with a residual policy trained through online RL, enabling improved performance without modifying the pre-trained diffusion model (Ankile et al., 2024).

## 3. Background

In this section, we offer fundamental definitions and theories to lay the groundwork for our framework, which will be thoroughly analyzed afterwards.

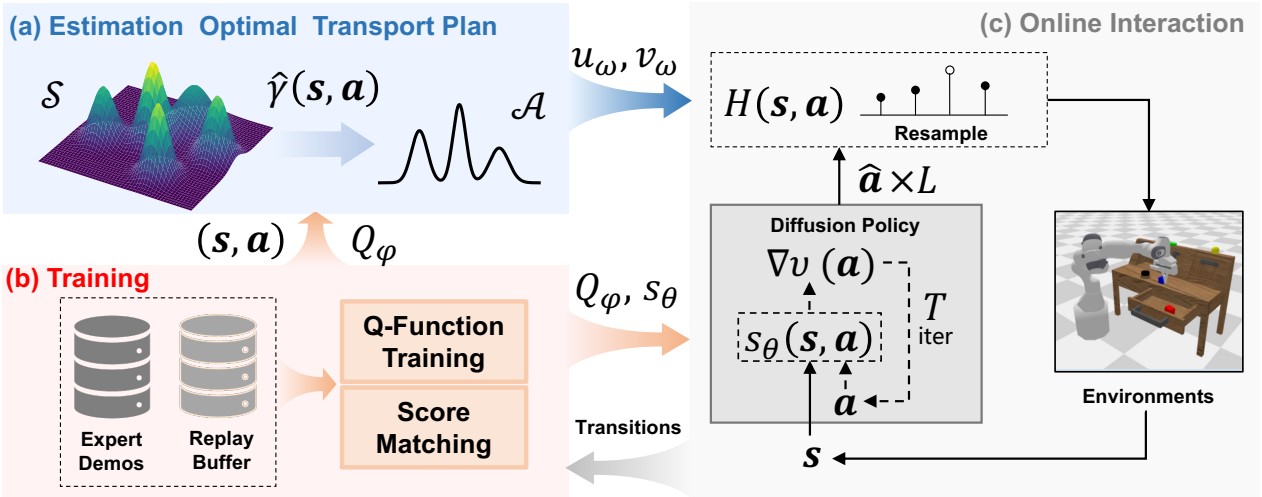

*Figure 1.* **Overview of OTPR.** (a) Estimation Optimal Transport Plan: A stochastic dual approach with two parametrized dual variables is introduced to estimate the optimal transport plan with $Q$-cost from state distribution and action distribution. (b) Training: OTPR pre-trains a diffusion model from the expert's data. It then iteratively performs RL to optimize a $Q$-function and trains diffusion models by score matching. (c) Online Interaction: In the inference step, the policy makes action inference by iteratively denoising a random noise, conditioned on current state. OTPR then employs the compatibility function $H$ to reweight each action before resampling one.

### 3.1. Optimal Transport

Given two probability spaces $(\mathcal{X}, \mu)$, $(\mathcal{Y}, \nu)$ and a cost function $c : \mathcal{X} \times \mathcal{Y} \rightarrow \mathbb{R}$, the Monge problem (Villani et al., 2009) is solving optimal map $\mathcal{T} : \mathcal{X} \rightarrow \mathcal{Y}$ such that

$$\inf \left\{ M(\mathcal{T}) := \mathbb{E}_{\boldsymbol{x} \sim \mu} \left[ c(\boldsymbol{x}, \mathcal{T}(\boldsymbol{x})) \right] \mid \mathcal{T}_{\#}\mu = \nu \right\} \quad (1)$$

where the random variables $\boldsymbol{x} \sim \mu$ and $\mathcal{T}_{\#}\mu$ is push-forward of $\mu$ subject to $(\mathcal{T}_{\#}\mu)(\mathcal{Y}') := \mu(\mathcal{T}^{-1}(\mathcal{Y}'))$ for any measurable set $\mathcal{Y}' \subset \mathcal{Y}$. Instead of finding the map $\mathcal{T}$ in the original Monge problem, the relaxed Kantorovich optimal scheme $K(\gamma)$ is obtained by $\gamma$ realizing

$$\inf \left\{ K(\gamma) := \mathbb{E}_{\boldsymbol{x} \times \boldsymbol{y} \sim \gamma}[c(\boldsymbol{x}, \boldsymbol{y})] \mid \gamma \in \Gamma(\mu, \nu) \right\}, \quad (2)$$

where $\Gamma(\mu, \nu)$ is the space composed of all joint probability measures $\gamma$ on $\mathcal{X} \times \mathcal{Y}$ with marginals $\mu$ and $\nu$.

**Regularized OT** Regularization was introduced in (Cuturi, 2013) to speed up the computation of OT problem, which is achieved by incorporating a negative-entropy penalty $R$ to the primal variable $\gamma$ of Problem 2,

$$\inf \left\{ K_\lambda(\gamma) := \mathbb{E}_{\boldsymbol{x} \times \boldsymbol{y} \sim \gamma}[c(\boldsymbol{x}, \boldsymbol{y})] + \lambda R(\gamma) \mid \gamma \in \Gamma(\mu, \nu) \right\}, \quad (3)$$

As highlighted by (Daniels et al., 2021), adding a regularization term with $\alpha$-strong convexity (such as entropy or squared $L_2$ norm) to the problem 3 is a sufficient condition for $\lambda\alpha$-strong convexity of $K_\lambda(\gamma)$ in $L_1$-norm, which makes the dual problem an unconstrained maximization problem.

In this work, we consider the $L_2$ regularization introduced by (Dessein et al., 2018), whose computation is found to be more stable since there is no exponential term causing overflow. For all $x \in \mathcal{X}$ and $y \in \mathcal{Y}$,

$$R_{L^2}(\gamma) \overset{\text{def.}}{=} \int_{\mathcal{X} \times \mathcal{Y}} \left( \frac{\mathrm{d}\gamma(x, y)}{\mathrm{d}\mu(x)\mathrm{d}\nu(y)} \right)^2 \mathrm{d}\mu(x)\mathrm{d}\nu(y). \quad (4)$$

where $\frac{\mathrm{d}\gamma(x,y)}{\mathrm{d}\mu(x)\mathrm{d}\nu(y)}$ is the density, i.e., the Radon-Nikodym derivative of $\gamma$ w.r.t. $\mu \times \nu$.

**Regularized OT Dual** We refer to the objective $K_\lambda(\gamma)$ as the primal objective, and we will use $\mathcal{J}_\lambda(u, v)$ to refer to the associated dual objective, with dual variables $u, v$. The dual of the regularized OT problems can be obtained through the Fenchel-Rockafellar's duality theorem,

$$\sup_{u,v} \mathbb{E}_{(\boldsymbol{x}, \boldsymbol{y}) \sim \mu \times \nu} \left[ u(\boldsymbol{x}) + v(\boldsymbol{y}) + F_\lambda(u(\boldsymbol{x}), v(\boldsymbol{y})) \right], \quad (5)$$

where $F_\lambda(u(\boldsymbol{x}), v(\boldsymbol{y})) = -\frac{1}{4\lambda}(u(\boldsymbol{x}) + v(\boldsymbol{y}) - c(\boldsymbol{x}, \boldsymbol{y}))_+^2$

is concave w.r.t. $(u, v)$ and $a_+ = \max(a, 0)$. In order to recover the solution $\gamma_\lambda$ of the regularized primal problem 3, we can use the first-order optimality conditions of the Fenchel-Rockafellar's duality theorem,

$$\mathrm{d}\gamma_\lambda(\boldsymbol{x}, \boldsymbol{y}) = H_\lambda(\boldsymbol{x}, \boldsymbol{y})\mathrm{d}\mu(\boldsymbol{x})\mathrm{d}\nu(\boldsymbol{y}) \quad (6)$$

where $H_\lambda(\boldsymbol{x}, \boldsymbol{y}) = \frac{1}{2\lambda}(u(\boldsymbol{x}) + v(\boldsymbol{y}) - c(\boldsymbol{x}, \boldsymbol{y}))_+.$

$H$ is called compatibility function.

## 3.2. Reinforcement Learning and Imitation Learning

**Reinfrocement Learning** We consider a standard Markov decision process (MDP) consisting of state space $s \in \mathcal{S}$, continuous action space $a \in \mathcal{A}$, deterministic state transition function $\mathcal{P} : \mathcal{S} \times \mathcal{A} \rightarrow \mathcal{S}$, reward function $r : \mathcal{S} \rightarrow \mathbb{R}$ and discount factor $\kappa$. $\tau \sim \pi$ denotes the distribution of trajectory $(s_0, a_0, s_1, a_1, \ldots)$ given the policy $\pi(a|s)$. The action-state value function is $Q_\pi(s, a) = \mathbb{E}_{\tau \sim \pi} [\sum_{t=0}^{\infty} \kappa^t r_t | a_0 = a, s_0 = s]$. The goal of RL is to learn the policy $\pi$ that maximizes the discounted expected cumulative reward over a trajectory $\tau$, defined as $\mathcal{J}_{RL}(\pi) = \mathbb{E}_{\tau \sim \pi}[\sum_{k=0} \kappa^k r_k]$.

**Imitation Learning** We assume access to a dataset $\mathcal{D}$ of demonstrations collected by expert human operators (often assumed to be optimal). Each trajectory $\tau \in \mathcal{D}$ consists of a sequence of transitions $\{(s_0, a_0), \ldots, (s_K, a_K)\}$. The most common IL method is behavior cloning (BC) which trains a parameterized policy $\pi_\theta$ to minimize the negative log-likelihood of data, i.e., $L(\theta) = -\mathbb{E}_{(s,a) \sim \mathcal{D}}[\log \pi_\theta(a|s)]$. In this work, we assume $\pi_\theta$ follows an isotropic Gaussian as its action distribution for simplicity. With the isotropic assumption, the BC training objective can be formulated as the following squared loss: $L^{IL}(\theta) = \mathbb{E}_{(s,a) \sim \mathcal{D}} \|\pi_\theta(s) - a\|_2^2$.

## 3.3. Conditional Score Based Diffusion Policy

The conditional Score Based Diffusion Models (SBDMs) (Song et al., 2021; Batzolis et al., 2021) aim to generate a target sample $y$ from the distribution $\mu$ of target training data given a condition data $x$. In imitation learning, diffusion policy regard state $s$ as condition $x$ and use a forward stochastic differential equation (SDE) to add Gaussian noises to the target training data $a$ for training the conditional score-based model. The forward SDE is $da_t = f(a_t, t)dt + g(t)dw$ with $a_0 \sim \nu$, where $w \in \mathbb{R}^D$ is a standard Wiener process, $f(\cdot, t) : \mathbb{R}^D \rightarrow \mathbb{R}^D$ is the drift coefficient, and $g(t) \in \mathbb{R}$ is the diffusion coefficient. Let $\nu_{t|0}$ be the conditional distribution of $a_t$ given the initial state $a_0$, and $\nu_t$ be the marginal distribution of $a_t$. The conditional score-based model is trained by denoising score-matching loss:

$$\mathcal{J}_{DSM}(\theta) = \mathbb{E}_t w_t \mathbb{E}_{a_0 \sim \nu} \mathbb{E}_{a_t \sim \nu_{t|0}(a_t|a_0)} \quad (7)$$
$$\left\| s_\theta(a_t; s_{\text{cond}}(a_0), t) - \nabla_{a_t} \log \nu_{t|0}(a_t|a_0) \right\|_2^2,$$

where $w_t$ is the weight for time $t$. In this paper, $t$ is uniformly sampled from $[0, T]$, i.e., $t \sim U([0, T])$. With the trained $s_{\hat{\theta}}(a; s, t)$, given a condition data $s$, the target sample $a_0$ is generated by the reverse SDE as $da_t = [f(a_t, t) - g(t)^2 s_{\hat{\theta}}(a_t; s, t)]dt + g(t)d\bar{w}$, where $\bar{w}$ is a standard Wiener process in the reverse-time direction.

## 4. Method

### 4.1. An Optimal Transport View of Policy Learning

We approach the policy optimization problem from the perspective of optimal transport. Considering Eq. 1, by substituting the cost function $c(x, a)$ with the critic $Q(s, a)$ and viewing our policy $\pi$ as a map that moves mass from the stationary state distribution $\mu(s)$, to the corresponding distribution of actions $\nu(a)$ given by an optimal behavior policy $\pi^\beta(\cdot|s)$, we formulate the following primal state-conditioned Monge OT problem:

$$\inf \left\{ M(\pi) := \mathbb{E}_{s \sim \mu} \left[ -Q^\beta(s, \pi(s)) \right] \mid \pi_\# \mu = \nu \right\} \quad (8)$$

The objective is to minimize the expectation of the negative critic function $Q^\beta$ while mapping exclusively to the distribution of actions given by the behavior policy $\pi^\beta$, a concept also explored in offline RL work (Asadulaev et al., 2025).

**Proposition 4.1.** *Given an optimal behavior policy $\pi^\beta$ and a critic-based cost function $c = -Q^\beta$, let $\pi^*$ is the solution to Eq. 8 with the $Q^\beta$ cost function. Then it holds that: $\mathcal{J}_{RL}(\pi^*) = \mathcal{J}_{RL}(\pi^\beta)$.*

The proof is given in Appendix B.1. Proposition 4.1 offers valuable insights into the connection between optimal transport theory and RL by establishing an equivalent relationship between the optimal transport map and the optimal policy. Meanwhile, given the paired state-action data derived from an expert policy, the IL problem can be reframed as achieving a conditional optimal transport map (i.e., diffusion policy). This link indicates that the transformations defined by the optimal transport map can effectively integrate RL with IL. We will next demonstrate how to use an estimated optimal transport plan (as intuitively illustrated in Fig. 4 of Appendix) to serve as a guide to utilize reinforcement learning to optimize the pre-trained diffusion policy via imitation learning.

### 4.2. OT-Guided Conditional Denoising Score Matching

In IL setting, we denote the all condition states as $s_{\text{cond}}(a)$ for a target action $a$, and $\mu$ is the measure by push-forwarding $\nu$ using $s_{\text{cond}}$, i.e., $\mu(s) = \sum_{\{a:s_{\text{cond}}(a)=s\}} \nu(a)$ over the paired training dataset $\mathcal{D}$. Section 3.3 provides a explicit reformulation for the conditional score-based diffusion policy with the paired training data.

**Proposition 4.2.** *Let $\mathcal{C}(s, a) = \frac{1}{\mu(s)} \delta_{s_{cond}(a)}(s)$ where $\delta$ is the generalized Dirac delta function, $\delta(s) = 0$, if $s \notin s_{cond}(a)$, then $\mathcal{J}_{DSM}(\theta)$ in Eq. 7 can be reformulated as*

$$\mathcal{J}_{CDSM}(\theta) = \mathbb{E}_t w_t \mathbb{E}_{s \sim \mu} \mathbb{E}_{a \sim \nu} \mathcal{C}(s, a) \mathbb{E}_{a_t \sim \nu_t|0}(a_t|a)$$
$$\left\| s_\theta(a_t; s, t) - \nabla_{a_t} \log \nu_{t|0}(a_t|a) \right\|_2^2. \quad (9)$$

*Furthermore, $\upsilon(s, a) = \mathcal{C}(s, a)\mu(s)\nu(a)$ is a joint distribution for marginal distributions $\mu$ and $\nu$.*

The proof is given in Appendix B.2. In Proposition 4.2, the coupling relationship of condition state and target action is explicitly modeled in $\mathcal{C}(\boldsymbol{s}, \boldsymbol{a})$. Nevertheless, in contrast to IL, the definition of $\mathcal{C}(\boldsymbol{s}, \boldsymbol{a})$ in RL is not explicit due to the absence of an optimal paired relationship between $\boldsymbol{s}, \boldsymbol{a}$. Fortunately, the joint distribution $\upsilon$ exhibits a similar formulation to the transport plan $\gamma$ in Eq. 6. We therefore use $L_2$-regularized OT to model the coupling relationship between state $\boldsymbol{s}$ and action $\boldsymbol{a}$ for unpaired settings. Specifically, given a $Q$ network learnt by RL, the $L_2$-regularized OT is applied to the distributions $\mu, \nu$ to approximately construct a conditional transport plan $\hat{\gamma}(\boldsymbol{a}|\boldsymbol{s}) = H(\boldsymbol{s}, \boldsymbol{a})\nu(\boldsymbol{a})$, and the coupling relationship of the condition data $\boldsymbol{s}$ and target data $\boldsymbol{a}$ is built by the compatibility function $H(\boldsymbol{s}, \boldsymbol{a})$. We then extend the formulation for paired setting in Eq. 9 by replacing $\mathcal{C}$ with $H$ to develop the training objective for unpaired setting, which is given by

$$\mathcal{J}_{\text{HDSM}}(\theta) = \mathbb{E}_t w_t \mathbb{E}_{\boldsymbol{s} \sim \mu} \mathbb{E}_{\boldsymbol{a} \sim \nu} H(\boldsymbol{s}, \boldsymbol{a}) \mathbb{E}_{\boldsymbol{a}_t \sim \nu_{t|0}}(\boldsymbol{a}_t | \boldsymbol{a})$$
$$\left\| s_\theta(\boldsymbol{a}_t; \boldsymbol{s}, t) - \nabla_{\boldsymbol{a}_t} \log \nu_{t|0}(\boldsymbol{a}_t | \boldsymbol{a}) \right\|_2^2. \quad (10)$$

In Eq. 10, $H$ is a "soft" coupling relationship of state data and action data, because there may exist multiple $a$ satisfying $H(\boldsymbol{s}, \boldsymbol{a}) > 0$ for each $s$. We minimize $\mathcal{J}_{\text{HDSM}}(\theta)$ to train the conditional score-based model $s_\theta(\boldsymbol{a}_t; \boldsymbol{s}, t)$.

**Theorem 4.3.** *For $\boldsymbol{s} \sim \mu$, consider the forward SDE $\mathrm{d}\boldsymbol{a}_t = f(\boldsymbol{a}_t, t)\mathrm{d}t + g(t)\mathrm{d}\mathbf{w}$ with $\boldsymbol{a}_0 \sim \hat{\gamma}(\cdot|\boldsymbol{s})$ and $t \in [0, T]$. Let $\nu_t(\boldsymbol{a}_t|\boldsymbol{s})$ be the distribution of $\boldsymbol{a}_t$ and $\mathcal{J}_{CSM}(\theta) = \mathbb{E}_t w_t \mathbb{E}_{\boldsymbol{s} \sim \mu} \mathbb{E}_{\boldsymbol{a}_t \sim \nu_t(\boldsymbol{a}_t|\boldsymbol{s})} \| s_\theta(\boldsymbol{a}_t; \boldsymbol{s}, t) - \nabla_{\boldsymbol{a}_t} \log \nu_t(\boldsymbol{a}_t|\boldsymbol{s}) \|_2^2$, then we have $\nabla_\theta \mathcal{J}_{HDSM}(\theta) = \nabla_\theta \mathcal{J}_{CSM}(\theta)$.*

We give the proof in Appendix B.3. Theorem 4.3 indicates that the trained $s_\theta(\boldsymbol{a}_t; \boldsymbol{s}, t)$ using Eq. 10 approximates $\nabla_{\boldsymbol{a}_t} \log \nu_t(\boldsymbol{a}_t|\boldsymbol{s})$. Based on this, we can interpret our approach as follows. Given a condition data $s$, we sample action data $\boldsymbol{a}_0$ from the conditional transport plan $\hat{\gamma}(\boldsymbol{a}_0|\boldsymbol{s})$, produce $\boldsymbol{a}_t$ by the forward SDE solver (examples given in Appendix A.2), and train $s_\theta(\boldsymbol{a}_t; \boldsymbol{s}, t)$ to approximate $\nabla_{\boldsymbol{a}_t} \log \nu_t(\boldsymbol{a}_t|\boldsymbol{s})$.

**Sample Generation** We denote the trained conditional score-based model as $s_{\hat{\theta}}(\boldsymbol{a}; \boldsymbol{s}, t)$ where $\hat{\theta}$ is the value of $\theta$ after training. Given the condition state $s$, we generate action samples by the following SDE:

$$\mathrm{d}\boldsymbol{a}_t = \left[f(\boldsymbol{a}_t, t) - g(t)^2 s_{\hat{\theta}}(\boldsymbol{a}; \boldsymbol{s}, t)\right] \mathrm{d}t + g(t)\mathrm{d}\bar{\mathbf{w}}. \quad (11)$$

Numerical solvers such as the Euler-Maruyama method, DDIM, or DPM-Solver can be employed to efficiently solve this reverse SDE, enabling the generation of high-quality action samples (Song et al., 2020a; Lu et al., 2022).

### 4.3. Expert Data Masked Optimal Transport

For the computation of $H$, a value-based reinforcement learning can provide an estimated Q-network, while opti-

mizing the optimal tranport problem gives $u, v$, which is often computationally challenging because OT needs transport all the mass of state to exactly match the mass of action distribution, which presents computational challenges. Fortunately, in imitation learning, expert demonstrations $\mathcal{D}^\beta$ have provide matched pairs of state and action data points (called "keypoints") $\mathcal{K} = \{(\boldsymbol{s}_i, \boldsymbol{a}_i)\}_{i=1}^N$. These keypoints are not only valuable but also crucial for investigating how to leverage them to guide the correct matching in OT. We introduce masked OT (Gu et al., 2022) to leverage the given matched keypoints to guide the correct transport in OT by preserving the relation of each data point to the keypoints:

$$\inf \left\{ K(\tilde{\gamma}) := \mathbb{E}_{\boldsymbol{s} \times \boldsymbol{a} \sim \tilde{\gamma}}[g(\boldsymbol{s}, \boldsymbol{a})] \,\Big|\, \tilde{\gamma} \in \tilde{\Gamma}(\mu, \nu; \mathbf{m}) \right\} \quad (12)$$

where the transport plan $\mathbf{m} \odot \tilde{\gamma}$ is $(\mathbf{m} \odot \tilde{\gamma})(\boldsymbol{s}, \boldsymbol{a}) = \mathbf{m}(s, a)\tilde{\gamma}(\boldsymbol{s}, \boldsymbol{a})$, and $\mathbf{m}$ is a binary mask function. Given a pair of keypoints $(\boldsymbol{s}_i, \boldsymbol{a}_i) \in \mathcal{K}$, then $\mathbf{m}(\boldsymbol{s}_i, \boldsymbol{a}_i) = 1, \mathbf{m}(\boldsymbol{s}_i, \boldsymbol{a}) = 0$ and $\mathbf{m}(\boldsymbol{s}, \boldsymbol{a}) = 1$ if $\boldsymbol{s}, \boldsymbol{a}$ do not coincide with any keypoint. The mask-based modeling of the transport plan ensures that the keypoint pairs are always matched in the derived transport plan. $g$ in Eq. 12 is defined as $g(\boldsymbol{s}, \boldsymbol{a}) = d(R_{\boldsymbol{s}}^s, R_{\boldsymbol{a}}^t)$, where $R_{\boldsymbol{s}}^s, R_{\boldsymbol{a}}^t \in (0, 1)^N$ model the vector of relation of $\boldsymbol{s}, \boldsymbol{a}$ to each of the paired keypoints in state and action space respectively, and $d$ is the Jensen–Shannon divergence. The $i$-th elements of $R_{\boldsymbol{s}}$ and $R_{\boldsymbol{a}}$ are respectively defined by

$$\begin{cases} R_{\boldsymbol{s},i}^s = \dfrac{\exp(-c(\boldsymbol{s}, s_i)/\rho)}{\sum_{j=1}^N \exp(-c(\boldsymbol{s}, s_j)/\rho)}, \\[3mm] R_{\boldsymbol{a},i}^t = \dfrac{\exp(-c(\boldsymbol{a}, a_i)/\rho)}{\sum_{j=1}^N \exp(-c(\boldsymbol{a}, a_j)/\rho)}, \end{cases} \quad (13)$$

where $\rho$ is a commonly used temperature in the softmax function. Further, the masked matrix is introduced into the duality of the $L^2$ reguarized OT problem, and the penalty term $F_\lambda$ is updated as:

$$\sup_{u, v} \mathbb{E}_{(\boldsymbol{s}, \boldsymbol{a}) \sim \mu \times \nu} \left[ u(\boldsymbol{s}) + v(\boldsymbol{a}) + \tilde{F}_\lambda(u(\boldsymbol{s}), v(\boldsymbol{a})) \right], \quad (14)$$

$$\tilde{F}_\lambda(u(\boldsymbol{s}), v(\boldsymbol{a})) = -\frac{1}{4\lambda} \mathbf{m}(\boldsymbol{s}, \boldsymbol{a})(u(\boldsymbol{s}) + v(\boldsymbol{a}) - g(\boldsymbol{s}, \boldsymbol{a}))_+^2.$$

The dual 5 and 14 are unconstrained concave, which can be maximized through stochastic gradient methods by sampling batches from $\mu \times \nu$. Following (Seguy et al., 2018), we use deep neural networks for their ability to approximate $u_\omega, v_\omega$ with the parameters $\omega$ and the estimate of OT plan is

$$\hat{\gamma}(\boldsymbol{s}, \boldsymbol{a}) = H(\boldsymbol{s}, \boldsymbol{a})\mathrm{d}\mu(\boldsymbol{s})\mathrm{d}\nu(\boldsymbol{a}), \quad (15)$$

$$\text{where } H(\boldsymbol{s}, \boldsymbol{a}) = \frac{1}{2\lambda}(u_\omega(\boldsymbol{s}) + v_\omega(\boldsymbol{a}) - g(\boldsymbol{s}, \boldsymbol{a}))_+.$$

The pseudo-codeis given in Appendix A.1.

**Algorithm 1** Online Score-Based Diffusion Policy Training

**Input:** The pre-trained imitation policy, expert demonstrations $\mathcal{D}$, initialzed $Q$-network and replay buffer $\mathcal{B}$.
**Output:** Trained conditional score-based policy $s_\theta$.
**for** iteration $= 1, 2, \ldots$ **do**
    Learn $u_\omega, v_\omega$ by optimizing the dual problem 14.
    **while** no done with episode **do**
        Observe current state $s$;
        Sample $a_l$ by the SDE 11, $l = 1, \ldots, L$;
        Compute $H(s, a_l)$ using Eq. 15;
        Normalize: $p_l = \frac{H(s, a_l)}{\sum_j H(s, a_j)}$;
        Select $a$ as a categorical from $p_l$;
        Store transition in $\mathcal{B}$.
    **end while**
    Learn $Q$-network by employed RL algorithm.
    Update $\theta$ of score model $s_\theta$ by fitting noise.
**end for**

### 4.4. OT-Guided Training

To implement $\mathcal{J}_{\text{HDSM}}(\theta)$ in Eq. 10 using training samples to optimize $\theta$, we can sample mini-batch data from replay buffer, and then compute $H(s, a)$ and $\mathcal{J}_{s,a} = \mathbb{E}_t w_t \mathbb{E}_{a_t \sim \nu_{t|0}(a_t|)} \|s_\theta(a_t; s, t) - \nabla_{a_t} \log \nu_{t|0}(a_t|a)\|_2^2$ over the pairs of $(s, a)$ in $\mathcal{S}$ and $\mathcal{A}$. However, such a strategy is sub-optimal. This is because given a mini-batch of samples $s$ and $a$, for each source sample $s$, there may not exist a target sample $a$ in the mini-batch with a higher value of $H(s, a)$ that matches condition data $s$. Therefore, few or even no samples in a mini-batch contribute to the loss function, leading to a large bias of the computed loss and instability of the training. To tackle this challenge, we generate $L$ samples from policy model, and then use the compatibility function to reweight these actions, ultimately forming the intended policy when resampled. This approach is summarized in Algorithm 1. In implementation, our approach can be used to replace the policy improvement step in multiple RL algorithms, while keeping the critic training as is. At evaluation time, we simply taking the action by setting $L = 1$ to reduce computational requirements.

## 5. Analysis

The proposed OTPR essentially aims to develop a conditional score-based diffusion policy for data transport from state space to action space in OT. To generate samples from conditional OT plan $\gamma^*(\cdot|s)$, the algorithm involves two key module learning: the dual term $(u_\omega, v_\omega)$ and the score model $s_\theta$. In this section, we will provide an analysis from the perspective of optimal transport, illustrating how the two aforementioned processes establish the upper bound of the distance between the distribution $\nu^{\text{SDE}}(a|s)$ of generated samples and the conditional optimal transport plan $\gamma^*(a|s)$.

To be specific, we investigate the upper bound of the expected Wasserstein distance $\mathbb{E}_{s \sim \mu} W_2(\nu^{\text{SDE}}(\cdot|s), \gamma^*(\cdot|s))$. Since $W_2(\cdot|\cdot)$ is a proper metric, we can conveniently leverage the triangle inequality to derive an upper bound for this expectation: $\mathbb{E}_{s \sim \mu} W_2(\nu^{\text{SDE}}(\cdot|s), \gamma^*(\cdot|s)) \leq \mathbb{E}_{s \sim \mu} W_2(\hat{\gamma}(\cdot|s), \gamma^*(\cdot|s)) + \mathbb{E}_{s \sim \mu} W_2(\nu^{\text{SDE}}(\cdot|s), \hat{\gamma}(\cdot|s))$, where $\hat{\gamma}$ is the estimated OT plan depending on $u_{\hat{\omega}}, v_{\hat{\omega}}$. This inequality provides a means to assess the upper bound by breaking it down into two more manageable comparisons.

To bound the first term, we denote the Lagrange function for $L_2$-regularized OTs in Eq. 5 as $\mathcal{L}(\gamma, u, v)$ with dual variables $u, v$ as follows:

$$
\begin{aligned}
\mathcal{L}(\gamma, u, v) = &\int \left( \gamma(s, a) + \lambda \frac{\gamma(s, a)^2}{\mu(s)\nu(s)} \right) \mathrm{d}s \, \mathrm{d}a \\
&+ \int u(s) \left( \int \gamma(s, a) \mathrm{d}a - \mu(s) \right) \mathrm{d}s \quad (16) \\
&+ \int v(a) \left( \int \gamma(s, a) \mathrm{d}s - \nu(a) \right) \mathrm{d}a.
\end{aligned}
$$

Because $\mathcal{L}(\gamma, u, v)$ is a sum of $K_\lambda(\gamma)$ and linear terms, the Lagrangian inherits $\lambda$-strong convexity in $L_1$-norm. Given the trained $u_{\hat{\omega}}$ and $v_{\hat{\omega}}$ which are $\epsilon$-approximate maximizers of $\mathcal{J}_\lambda(u, v)$, the pseudo-plan $\hat{\gamma} = H(s, a; u_{\hat{\omega}}, v_{\hat{\omega}})\mu(s)\nu(a)$ satisfies:

$$
\frac{\lambda}{2} \|\hat{\gamma} - \gamma^*\|_1^2 \leq \mathcal{L}(\hat{\gamma}, u_{\hat{\omega}}, v_{\hat{\omega}}) - \mathcal{L}(\gamma^*, u^*, v^*) \leq \epsilon \quad (17)
$$

Since the strong convexity of $\mathcal{L}$ implies a Polyak-Łojasiewicz (PL) inequality, we have,

$$
\|\hat{\gamma} - \gamma^*\|_1 \leq \frac{1}{\lambda} \|\nabla_{\hat{\gamma}} L(\hat{\gamma}, u_{\hat{\omega}}, v_{\hat{\omega}})\|_1 \quad (18)
$$

Consequently, we can derive an upper bound for the expected Wasserstein distance as follows:

$$
\mathbb{E}_{s \sim \mu} W_2(\hat{\gamma}(\cdot|s), \gamma^*(\cdot|s)) \leq \frac{\eta}{\lambda} \|\nabla_{\hat{\gamma}} L(\hat{\gamma}, u_{\hat{\omega}}, v_{\hat{\omega}})\|_1, \quad (19)
$$

where $\eta = \max_{a, a' \in \mathcal{A}} \{\|a - a'\|_2\}$.

For the bound of $\mathbb{E}_{s \sim \mu} W_2(\nu^{\text{SDE}}(\cdot|s), \hat{\gamma}(\cdot|s))$, it is difficult to get without explicit $f$ and $g$ given, but from the existing convergence guarantees for a general class of score-based generative models, we get $\mathbb{E}_{s \sim \mu} W_2(\nu^{\text{SDE}}(\cdot|s), \hat{\gamma}(\cdot|s)) \leq \epsilon$, which can be easily interpreted as two terms (1) the initialization of the algorithm at $\hat{\nu}_T(\cdot|s)$ instead of $\hat{\gamma}_T((\cdot|s))$, (2) the discretization and score-matching errors in running the algorithm (Kwon et al., 2022; Gao et al., 2023).

## 6. Experiments

In this section, we evaluate OTPR and several prior approaches, in a number of benchmark domains that require learning policies from static offline expert data and then

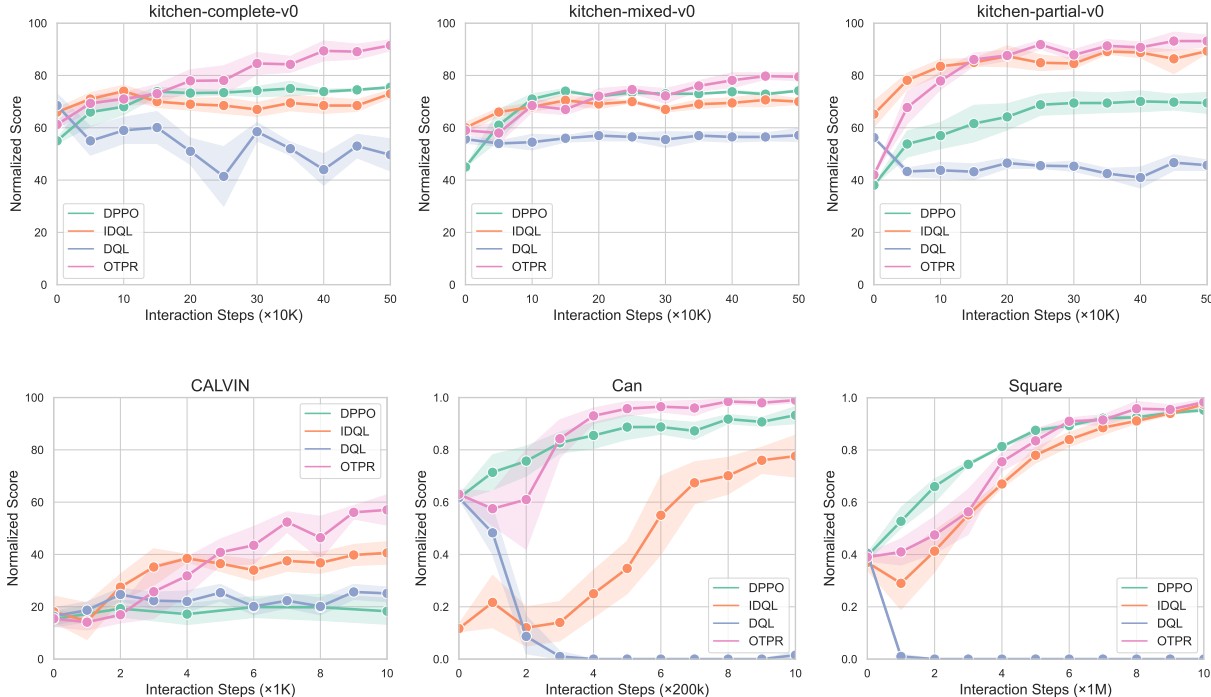

Figure 2. **Learning curves of online fine-tuning with various methods.** Observe that OTPR largely always dominates or attains similar performance to the next best method. Other methods for fine-tuning diffusion policies (IDQL, DQL, DPPO) are a bit unstable, and perform substantially worse.

fine-tune them with limited online interaction in the MDP (offline-to-online fine-tuning). We also study the hybrid RL problem setting (i.e., online RL with offline data put in the replay buffer) for some experiments. More results are shown on Appendix D. Finally, we perform ablation experiments to understand the utility of different components of OTPR.

### 6.1. Experimental Setup

**Environments and tasks.** We study: (1) Robomimic tasks (Mandlekar et al., 2021), which is a commonly used benchmark designed to study imitation learning for robot manipulation.The evaluation score represents the success rate. (2) Franka-Kitchen tasks (Gupta et al., 2019), which require solving a sequence of four manipulation tasks in a kitchen environment with a 9-Dof Franka robot; and (3) the CALVIN benchmark (Mees et al., 2022), an evaluation benchmark designed for long-horizon, language-conditioned manipulation, which requires solving a sequence of four manipulation tasks in a tabletop environment. The evaluation score for a trajectory is the maximum number of sub-tasks completed simultaneously at any single point in the trajectory. The CALVIN task is significantly challenging, as policies must be learned directly from pixels using offline play data obtained through human teleoperation.

**Implementation details.** We provide a detailed list of

hyper-parameters and best practices for running OTPR in Appendix. We instantiate OTPR using the popular IQL with keeping the critic training as is. For the image-based domain, we use a ResNet 18 encoder and store features in the replay buffer to facilitate the estimation of the dual terms.

### 6.2. Results

**Comparisons with Other Online Fine-Tuning Methods.** We conduct a comprehensive comparison of OTPR against a range of reinforcement learning (RL) methods designed for fine-tuning diffusion-based policies. Specifically, we evaluate the following approaches: (1) Implicit Diffusion Q-Learning (IDQL) (Hansen-Estruch et al., 2023), which extends Implicit Q-Learning (IQL) to incorporate diffusion policies through critic-based re-ranking; (2) Diffusion Policy Optimization (DPPO) (Ren et al., 2024), which fine-tunes diffusion policies initially learned via imitation learning by optimizing a two-layer Markov Decision Process (MDP) loss; and (3) Diffusion Q-Learning (DQL) (Wang et al., 2022), which trains diffusion policies using a reparameterized policy gradient estimator similar to the Soft Actor-Critic (SAC) framework (Haarnoja et al., 2018).

Overall, OTPR performs consistently and significantly improves fine-tuning efficiency and asymptotic performance of diffusion policies. Notably, OTPR consistently main-

*Table 1.* Comparison of OTPR with other demo-augmented RL algorithms. OTPR outperforms every other approach, both in terms of the offline performance (left of →) and performance after fine-tuning (right of →).

| | Franka-Kitchen | | | RoboMimic | |
| | Kitchen-Complete-v0 | Kitchen-Mixed-v0 | Kitchen-Partial-v0 | Can-State | Square-State |
|---|---|---|---|---|---|
| RLPD | $0 \rightarrow 18$ | $0 \rightarrow 14$ | $0 \rightarrow 34$ | $0 \rightarrow 0$ | $0 \rightarrow 3$ |
| Cal-QL | $19 \rightarrow 57$ | $37 \rightarrow 72$ | $59 \rightarrow 84$ | $0 \rightarrow 0$ | $0 \rightarrow 0$ |
| IBRL | $0 \rightarrow 25$ | $0 \rightarrow 13$ | $0 \rightarrow 15$ | $0 \rightarrow 64$ | $0 \rightarrow 50$ |
| OTPR | $61 \rightarrow 92$ | $59 \rightarrow 79$ | $42 \rightarrow 93$ | $63 \rightarrow 99$ | $40 \rightarrow 98$ |

tains high normalized scores in the kitchen-complete-v0, CALVIN and Can task, while other methods exhibit relative instability, especially DQL and IDQL, which show considerable fluctuations in performance across different interaction steps. This may be attributed to both DQL and IDQL performing off-policy updates and propagating gradients from the imperfect Q function to the actor, which results in even greater training instability in sparse-reward tasks given the continuous action space and large action chunk sizes. In contrast, OTPR can quickly mitigate the adverse effects brought about by this issue by leveraging the guidance of the compatible function. This analysis suggests that OTPR is a robust and effective approach for online fine-tuning in diffusion policy tasks, consistently outperforming the other methods in terms of stability and overall performance.

**Comparisons with demo-augmented RL.** Next, we compare OTPR with recently proposed RL methods for training robot policies (not necessarily diffusion based) leveraging offline data, including RLPD (Ball et al., 2023), Cal-QL (Nakamoto et al., 2024), and IBRL (Hu et al., 2023). These methods add expert data in the replay buffer and performs off-policy updates. We evaluate these methods on Franka-Kitchen and RoboMimic environents. IBRL and Cal-QL are also pretrained with behavior cloning and offline RL objectives, respectively. All of results are shown on Table 1. In the Franka-Kitchen domains, while Cal-QL demonstrates competitive performance, OTPR shows more impressive score improvements, rising from 61 to 92 in Kitchen-Complete-v0 and from 59 to 79 in Kitchen-Mixed-v0. In contrast, other methods such as RLPD, IQL, and IBRL perform significantly worse, particularly in the Kitchen-Partial-v0 task, where OTPR leads with a final score of 93. In the RoboMimic environment, OTPR continues to excel, achieving high scores of 99 in Can-State and 98 in Square-State, showcasing its robustness across diverse scenarios. Although IBRL performs the best among the competitors, there remains a significant gap in performance.

### 6.3. Ablation Experiments

**Effect of the compatibility function.** In the previous section, we have already demonstrated the advantages of OTPR

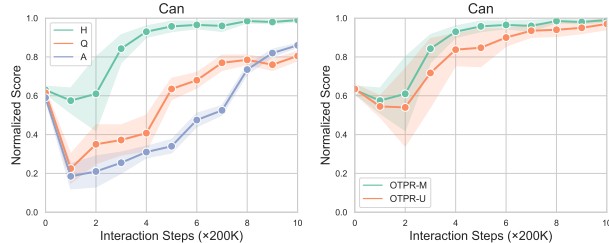

*Figure 3.* (left) Comparison between OTPR with different guidance (H, Q and A). (right) Comparison between OTPR with (OTPR-M) and without (OTPR-U) the expert data mask.

over other diffusion-based fine-tuning methods that rely on Q-values. Now, to spotlight the pivotal role of our method's core component—the guidance from the compatibility function $H$, we replace it with $Q$ and advantages $A$ within the same training framework. The experimental results on the Robomimic-Can task are illustrated in Fig. 3(left). Clearly, compared to using Q-value and advantage, OT-guided training demonstrates significantly faster convergence and superior evaluation performance.

**Effect of the masked OT.** OTPR incorporates masked Optimal Transport (OT) to utilize expert data as keypoints, guiding accurate distribution transport. As depicted in Fig. 3, OTPR-U, which lacks the mask matrix, exhibits instability and reduced efficiency, despite outperforming other mainstream methods. Notably, even without the mask, OTPR can still operate as a fully functional offline RL algorithm by leveraging the compatibility function without reward.

## 7. Conclusion

This paper introduced OTPR, a novel method integrating optimal transport theory with diffusion policies to enhance the efficiency and adaptability of reinforcement learning fine-tuning. OTPR leverages the Q-function as a transport cost and uses masked optimal transport to guide state-action matching, improving learning stability and performance. Experiments demonstrated OTPR's superior performance across multiple tasks, especially in complex environments.

Future work will focus on scaling OTPR to larger state-action spaces, and exploring its integration with other advanced policy architectures.

## Acknowledgements

This work was supported by the National Science and Technology Innovation 2030 - Major Project (Grant No. 2022ZD0208800), and NSFC General Program (Grant No. 62176215).

## Impact Statement

This paper presents work whose goal is to advance the field of reinforcement learning for robotic application. There are many potential societal consequences of our work, none of which we feel must be specifically highlighted here.

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

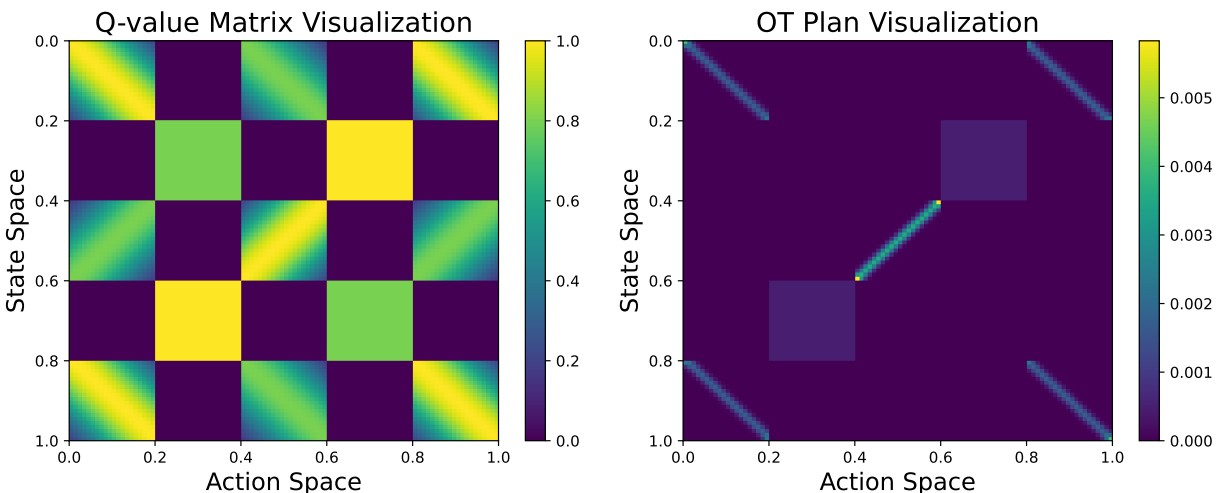

*Figure 4.* This example demonstrates a clear and concise visualization of a Q-value matrix alongside its corresponding estimated optimal transport plan.

## A. Additional Details for Algorithm

### A.1. Pseudo-codes of algorithm for training $u_\omega, v_\omega$.

The pseudo-codes of the algorithm to learn the dual terms $u_\omega, v_\omega$, a.k.a., potentials, are given in Algorithm 2 (Seguy et al., 2018; Gu et al., 2023).

---

**Algorithm 2** Algorithm for estimating potentials $u_{\omega_1}, v_{\omega_2}$

**Input:** $Q$-network and replay buffer $\mathcal{B}$, nets $u_\omega, v_\omega$, batch size $b$, learning rate $\alpha$, expert demonstrations $\mathcal{D}$ (if available) .
**Output:** Learned potential $u_{\omega_1}, v_{\omega_2}$.
**for** iteration $= 1, 2, \ldots$ **do**
    Calculate the cost between $\boldsymbol{s}$ and $\boldsymbol{a}$ in $\mathcal{B}$.
    Sample a state batch $\{s_1, s_2, \ldots, s_b\}$ from $\mathcal{B}$.
    Sample a action batch $\{a_1, a_2, \ldots, a_b\}$ from $\mathcal{B}$.
    **if** Expert data $\mathcal{D}$ is available **then**
        Update $\omega_1 \leftarrow \omega_1 + \alpha \sum_{i,j} \nabla_{\omega_1} u_{\omega_1}(s_i) + \partial_u \tilde{F}_\lambda(u_{\omega_1}(s_i), v_{\omega_2}(a_j)) \nabla_{\omega_1} u_{\omega_1}(s_i)$
        Update $\omega_2 \leftarrow \omega_2 + \alpha \sum_{i,j} \nabla_{\omega_2} v_{\omega_2}(a_j) + \partial_v \tilde{F}_\lambda(u_{\omega_1}(s_i), v_{\omega_2}(a_j)) \nabla_{\omega_2} v_{\omega_2}(a_m)$
    **else**
        Update $\omega_1 \leftarrow \omega_1 + \alpha \sum_{i,j} \nabla_{\omega_1} u_{\omega_1}(s_i) + \partial_u F_\lambda(u_{\omega_1}(s_i), v_{\omega_2}(a_j)) \nabla_{\omega_1} u_{\omega_1}(s_i)$
        Update $\omega_2 \leftarrow \omega_2 + \alpha \sum_{i,j} \nabla_{\omega_2} v_{\omega_2}(a_j) + \partial_v F_\lambda(u_{\omega_1}(s_i), v_{\omega_2}(a_j)) \nabla_{\omega_2} v_{\omega_2}(a_m)$
    **end if**
**end for**

---

### A.2. Training by fitting noise.

We consider the VE-SDE and the VP-SDE as examples of forward SDEs. In the VE-SDE, $f(\boldsymbol{a}, t) = 0$ and $g(t) = \sqrt{\frac{d[\sigma^2(t)]}{dt}}$, where $\sigma > 0$ is an increasing function of t. For the VP-SDE, $f(\boldsymbol{a}, t) = -\frac{1}{2}\beta(t)\boldsymbol{a}$ and $g(t) = \sqrt{\beta(t)}$, with $\beta(t) = \beta_{min} + (\beta_{max} - \beta_{min})t$. The distribution $p_{t|0}(\boldsymbol{a}_t|\boldsymbol{a}_0)$ for $\boldsymbol{a}_t$ given $\boldsymbol{a}_0$ is defined as:

$$p_{t|0}(\boldsymbol{a}_t|\boldsymbol{a}_0) = \begin{cases} \mathcal{N}(\boldsymbol{a}_t|\boldsymbol{a}_0, \sigma^2(t)\mathbf{I}), & \text{for VE-SDE,} \\ \mathcal{N}(\boldsymbol{a}_t|\boldsymbol{a}_0 e^{\frac{1}{2}h(t)}, (1 - e^{h(t)})\mathbf{I}), & \text{for VP-SDE,} \end{cases} \tag{20}$$

where $h(t) = -\frac{1}{2}t^2(\beta_{max} - \beta_{min}) - t\beta_{min}$, and $\mathbf{I}$ is the identity matrix.

We define the $\sigma_t \mathbf{I}$ as the standard variation of $p_{t|0}(\boldsymbol{a}_t|\boldsymbol{a})$, specifically, $\sigma_t^2 = \sigma^2(t)$ for VE-SDE and $\sigma_t^2 = 1 - e^{h(t)}$ for VP-SDE. Using the reparameterization trick, given sampling$(\boldsymbol{s}, \boldsymbol{a})$, we have $\boldsymbol{a}_t = \boldsymbol{a} + \sigma_t \epsilon$ for VE-SDE, and $\boldsymbol{a}_t = e^{\frac{1}{2}h(t)}\boldsymbol{a} + \sigma_t \epsilon$ for VP-SDE, where $\epsilon \sim \mathcal{N}(0, \mathbf{I})$. Further, $\nabla_{\boldsymbol{a}_t} \log p_{t|0}(\boldsymbol{a}_t|\boldsymbol{a}) = -\frac{1}{\sigma_t}\epsilon$. Therefore, the loss $\mathcal{J}_{\boldsymbol{s},\boldsymbol{a}}$ for fitting noise can be written as

$$\mathcal{J}_{\boldsymbol{s},\boldsymbol{a}} = \mathbb{E}_{t,\epsilon \sim \mathcal{N}(0,I)} \left[ \frac{w_t}{\sigma_t^2} \left\| s_\theta(\nu_t(\boldsymbol{a}) + \sigma_t \epsilon; \boldsymbol{s}, t)\sigma_t + \epsilon \right\|_2^2 \right]. \tag{21}$$

For VE-SDE, $\nu_t(\boldsymbol{a}) = \boldsymbol{a}$, while for VP-SDE, $\nu_t(\boldsymbol{a}) = e^{\frac{1}{2}h(t)}\boldsymbol{a}$. Equation 21 indicates that $s_\theta(y_t; x, t)$ is trained to match the scaled noise $-\frac{1}{\sigma_t}\epsilon$.

## B. Proofs

### B.1. Proof of Proposition 4.1

**Proposition B.1.** *Given an optimal behavior policy $\pi^\beta$ and a critic-based cost function $c = -Q^\beta$, let $\pi^*$ is the solution to Eq. 8 with the $Q^\beta$ cost function. Then it holds that: $\mathcal{J}_{RL}(\pi^*) = \mathcal{J}_{RL}(\pi^\beta)$.*

*Proof.* We use Supp$(\mu)$ and Supp$(\nu)$ to refer to the support of $\mu$ and $\nu$, two subsets of $\mathcal{S}$ and $\mathcal{A}$, respectively, which are also the set of values which $\boldsymbol{s} \sim \mu$ and $\boldsymbol{a} \sim \nu$ can take. Given a point $s \in$ Supp$(\mu)$, the Monge problem would send the whole mass at $x$ to a unique location $a \in$ Supp$(\nu)$. The a primal state-conditioned Monge OT problem discripted with 8 can be formulated as:

$$\inf_\pi \mathbb{E}_{\boldsymbol{s} \sim \mu} \left[ -Q^\beta(\boldsymbol{s}, \pi(\boldsymbol{s})) \right], \text{subject to } \pi(s) \in \text{Supp}(\nu) \text{ for all } s \in \text{Supp}(\mu). \tag{22}$$

According to Lemma 6.1 in (Kakade & Langford, 2002), we can use the performance difference lemma to compare the performance of the two policies $\pi^*$ and $\pi^\beta$ :

$$J(\pi^*) - J(\pi^\beta) = \frac{1}{1 - \kappa}\mathbb{E}_{\boldsymbol{s} \sim \mu} \left[ A^\beta(\boldsymbol{s}, \pi^*) \right] \tag{23}$$

$$= \frac{1}{1 - \kappa}\mathbb{E}_{\boldsymbol{s} \sim \mu} \left[ Q^\beta(\boldsymbol{s}, \pi^*(\boldsymbol{s})) - V^\beta(\boldsymbol{s}) \right] \tag{24}$$

$$= \frac{1}{1 - \kappa}\mathbb{E}_{\boldsymbol{s} \sim \mu} \left[ Q^\beta(\boldsymbol{s}, \max_{\boldsymbol{a} \in \text{Supp}(\beta(\cdot|\boldsymbol{s}))}[Q^\beta(\boldsymbol{s}, \boldsymbol{a})]) - V^\beta(\boldsymbol{s}) \right] \tag{25}$$

In the setting of Proposition, $\pi^\beta$ is an optimal expert policy, $V^\beta(\boldsymbol{s}) = \max_{\boldsymbol{a}} Q^\beta(\boldsymbol{s}, \boldsymbol{a})$, thus Supp$(\pi^\beta(\cdot|s))$ indicates the optimal actions $\boldsymbol{a}^*$ from $\pi^\beta$ which maximize $Q^\beta$. Then we have: $J(\pi^*) - J(\pi^\beta) = 0$. $\square$

### B.2. Proof of Proposition 4.2

**Proposition B.2.** *Let $\mathcal{C}(\boldsymbol{s}, \boldsymbol{a}) = \frac{1}{\mu(\boldsymbol{s})}\delta(\boldsymbol{s} - \boldsymbol{s}_{cond}(\boldsymbol{a}))$ where $\delta$ is the Dirac delta function, then $\mathcal{J}_{DSM}(\theta)$ in Eq. 7 can be reformulated as*

$$\mathcal{J}_{CDSM}(\theta) = \mathbb{E}_t w_t \mathbb{E}_{\boldsymbol{s} \sim \mu} \mathbb{E}_{\boldsymbol{a} \sim \nu} \mathcal{C}(\boldsymbol{s}, \boldsymbol{a}) \mathbb{E}_{\boldsymbol{a}_t \sim \nu_{t|0}}(\boldsymbol{a}_t|\boldsymbol{a}) \\ \left\| s_\theta(\boldsymbol{a}_t; \boldsymbol{s}, t) - \nabla_{\boldsymbol{a}_t} \log \nu_{t|0}(\boldsymbol{a}_t|\boldsymbol{a}) \right\|_2^2. \tag{26}$$

*Furthermore,*

$$\upsilon(\boldsymbol{s}, \boldsymbol{a}) = \mathcal{C}(\boldsymbol{s}, \boldsymbol{a})\mu(\boldsymbol{s})\nu(\boldsymbol{a}) \tag{27}$$

*is a joint distribution for marginal distributions $\mu$ and $\nu$.*

*Proof.* We first prove Eq. 26, and then demonstrate that $\upsilon(\boldsymbol{s}, \boldsymbol{a})$ serves as a joint distribution for the marginal distributions $\mu$ and $\nu$. (1) The right side of Eq. 26 is

$$\mathbb{E}_t w_t \mathbb{E}_{\boldsymbol{s}\sim\mu} \mathbb{E}_{\boldsymbol{a}\sim\nu} \mathcal{C}(\boldsymbol{s}, \boldsymbol{a}) \mathbb{E}_{\boldsymbol{a}_t\sim\nu_{t|0}(\boldsymbol{a}_t|\boldsymbol{a})} \|s_\theta(\boldsymbol{a}_t; \boldsymbol{s}, t) - \nabla_{\boldsymbol{a}_t} \log \nu_{t|0}(\boldsymbol{a}_t|\boldsymbol{a})\|_2^2 \tag{28}$$

$$=\mathbb{E}_t w_t \mathbb{E}_{\boldsymbol{a}\sim\nu} \int \mu(\boldsymbol{s}) \mathcal{C}(\boldsymbol{s}, \boldsymbol{a}) \mathbb{E}_{\boldsymbol{a}_t\sim\nu_{t|0}(\boldsymbol{a}_t|\boldsymbol{a})} \|s_\theta(\boldsymbol{a}_t; \boldsymbol{s}, t) - \nabla_{\boldsymbol{a}_t} \log \nu_{t|0}(\boldsymbol{a}_t|\boldsymbol{a})\|_2^2 d\boldsymbol{s} \tag{29}$$

$$=\mathbb{E}_t w_t \mathbb{E}_{\boldsymbol{a}\sim\nu} \int \delta(\boldsymbol{s} - \boldsymbol{s}_{\text{cond}}(\boldsymbol{a})) \mathbb{E}_{\boldsymbol{a}_t\sim\nu_{t|0}(\boldsymbol{a}_t|\boldsymbol{a})} \|s_\theta(\boldsymbol{a}_t; \boldsymbol{s}, t) - \nabla_{\boldsymbol{a}_t} \log \nu_{t|0}(\boldsymbol{a}_t|\boldsymbol{a})\|_2^2 d\boldsymbol{s} \tag{30}$$

$$=\mathbb{E}_t w_t \mathbb{E}_{\boldsymbol{a}\sim\nu} \mathbb{E}_{\boldsymbol{a}_t\sim\mu_{t|0}(\boldsymbol{a}_t|\boldsymbol{a})} \left\| s_\theta(\boldsymbol{a}_t; \boldsymbol{s}_{\text{cond}}(\boldsymbol{a}), t) - \nabla_{\boldsymbol{a}_t} \log \nu_{t|0}(\boldsymbol{a}_t|\boldsymbol{a}) \right\|_2^2, \tag{31}$$

which is the definition of $\mathcal{J}_{\text{DSM}}(\theta)$ in Eq. 7.

(2) We demonstrate that the marginal distributions of $\upsilon(\boldsymbol{s}, \boldsymbol{a})$ are $\mu$ and $\nu$ as follows. Firstly,

$$\int \upsilon(\boldsymbol{s}, \boldsymbol{a}) \, d\boldsymbol{s} = \int \delta(\boldsymbol{s} - \boldsymbol{s}_{\text{cond}}(\boldsymbol{a})) \nu(\boldsymbol{a}) \, d\boldsymbol{s} = \nu(\boldsymbol{a}) \int \delta(\boldsymbol{s} - \boldsymbol{s}_{\text{cond}}(\boldsymbol{a})) \, d\boldsymbol{s} = \nu(\boldsymbol{a}) \tag{32}$$

Next, from the definition of $\delta(\cdot)$, we obtain $\delta(\boldsymbol{s} - \boldsymbol{s}_{\text{cond}}(\boldsymbol{a})) = \sum_{\boldsymbol{a}':\boldsymbol{s}_{\text{cond}}(\boldsymbol{a}')=\boldsymbol{s}} \delta(\boldsymbol{a} - \boldsymbol{a}')$. Then, we have

$$\int \upsilon(\boldsymbol{s}, \boldsymbol{a}) \, d\boldsymbol{a} = \int \delta(\boldsymbol{s} - \boldsymbol{s}_{\text{cond}}(\boldsymbol{a})) \nu(\boldsymbol{a}) \, d\boldsymbol{a} \tag{33}$$

$$= \int \sum_{\{\boldsymbol{a}':\boldsymbol{s}_{\text{cond}}(\boldsymbol{a}')=\boldsymbol{s}\}} \delta(\boldsymbol{a}' - \boldsymbol{a}) \nu(\boldsymbol{a}) \, d\boldsymbol{a} \tag{34}$$

$$= \sum_{\{\boldsymbol{a}':\boldsymbol{s}_{\text{cond}}(\boldsymbol{a}')=\boldsymbol{s}\}} \int \delta(\boldsymbol{a}' - \boldsymbol{a}) \nu(\boldsymbol{a}) \, d\boldsymbol{a} \tag{35}$$

$$= \sum_{\{\boldsymbol{a}':\boldsymbol{s}_{\text{cond}}(\boldsymbol{a}')=\boldsymbol{s}\}} \nu(\boldsymbol{a}') \tag{36}$$

$$= \mu(\boldsymbol{s}) \tag{37}$$

$\square$

## B.3. Proof of Theorem 4.3

**Theorem B.3.** *For $\boldsymbol{s} \sim \mu$, consider the forward SDE $d\boldsymbol{a}_t = f(\boldsymbol{a}_t, t)dt + g(t)d\mathbf{w}$ with $\boldsymbol{a}_0 \sim \hat{\gamma}(\cdot|\boldsymbol{s})$ and $t \in [0, T]$. Let $\nu_t(\boldsymbol{a}_t|\boldsymbol{s})$ be the distribution of $\boldsymbol{a}_t$ and $\mathcal{J}_{\text{CSM}}(\theta) = \mathbb{E}_t w_t \mathbb{E}_{\boldsymbol{s}\sim\mu} \mathbb{E}_{\boldsymbol{a}_t\sim\nu_t(\boldsymbol{a}_t|\boldsymbol{s})} \|s_\theta(\boldsymbol{a}_t; \boldsymbol{s}, t) - \nabla_{\boldsymbol{a}_t} \log \nu_t(\boldsymbol{a}_t|\boldsymbol{s})\|_2^2$, then we have $\nabla_\theta \mathcal{J}_{\text{HDSM}}(\theta) = \nabla_\theta \mathcal{J}_{\text{CSM}}(\theta)$.*

*Proof.* To establish the equivalence between $\mathcal{J}_{\text{HDSM}}(\theta)$ and $\mathcal{J}_{\text{CSM}}(\theta)$, we start by examining the difference between the two objective functions:

$$\mathcal{J}_{\text{HDSM}}(\theta) - \mathcal{J}_{\text{CSM}}(\theta) = \mathbb{E} w_t \mathbb{E}_{\boldsymbol{s}\sim\mu} \mathbb{E}_{\boldsymbol{a}_0\sim\nu} H(\boldsymbol{s}, \boldsymbol{a}_0) \mathbb{E}_{\boldsymbol{a}_t\sim\nu_{t|0}(\boldsymbol{a}_t|\boldsymbol{a}_0)} \|s_\theta(\boldsymbol{a}_t; \boldsymbol{s}, t) - \nabla_{\boldsymbol{a}_t} \log \nu_{t|0}(\boldsymbol{a}_t|\boldsymbol{a}_0)\|_2^2 - \mathcal{J}_{\text{CSM}}(\theta)$$

$$= \mathbb{E} w_t \mathbb{E}_{\boldsymbol{s}\sim\mu} \mathbb{E}_{\boldsymbol{a}_0\sim\hat{\gamma}(\boldsymbol{a}_0|\boldsymbol{s})} \mathbb{E}_{\boldsymbol{a}_t\sim\nu_{t|0}(\boldsymbol{a}_t|\boldsymbol{a}_0)} \|s_\theta(\boldsymbol{a}_t; \boldsymbol{s}, t) - \nabla_{\boldsymbol{a}_t} \log \nu_{t|0}(\boldsymbol{a}_t|\boldsymbol{a}_0)\|_2^2 - \mathcal{J}_{\text{CSM}}(\theta) \tag{38}$$

Since $\boldsymbol{s} \to \boldsymbol{a}_0 \to \boldsymbol{a}_t$ is a Markov Chain in the forward SDE process, the distribution $\nu_{t|0}(\boldsymbol{a}_t|\boldsymbol{a}_0, \boldsymbol{s})$ of $\boldsymbol{a}_t$ simplifies to $\nu_{t|0}(\boldsymbol{a}_t|\boldsymbol{a}_0, \boldsymbol{s}) = \nu_{t|0}(\boldsymbol{a}_t|\boldsymbol{a}_0)$, which is the distribution of $\boldsymbol{a}_t$ by the forward SDE $d\boldsymbol{a}_t = f(\boldsymbol{a}_t, t)dt + g(t)d\mathbf{w}$ with initial state $\boldsymbol{a}_0$. Then, we have

$$\mathcal{J}_{\text{HDSM}}(\theta) - \mathcal{J}_{\text{CSM}}(\theta) = \mathbb{E} w_t \mathbb{E}_{\boldsymbol{s}\sim\mu} \mathbb{E}_{\boldsymbol{a}_0\sim\hat{\gamma}(\boldsymbol{a}_0|\boldsymbol{s})} \mathbb{E}_{\boldsymbol{a}_t\sim\nu_{t|0}(\boldsymbol{a}_t|\boldsymbol{a}_0,\boldsymbol{s})} \|s_\theta(\boldsymbol{a}_t; \boldsymbol{s}, t) - \nabla_{\boldsymbol{a}_t} \log \nu_{t|0}(\boldsymbol{a}_t|\boldsymbol{a}_0, \boldsymbol{s})\|_2^2$$

$$- \mathbb{E}_t w_t \mathbb{E}_{\boldsymbol{s}\sim\mu} \mathbb{E}_{\boldsymbol{a}_t\sim\nu_t(\boldsymbol{a}_t|\boldsymbol{s})} \|s_\theta(\boldsymbol{a}_t; \boldsymbol{s}, t) - \nabla_{\boldsymbol{a}_t} \log \nu_t(\boldsymbol{a}_t|\boldsymbol{s})\|_2^2. \tag{39}$$

According to (Vincent, 2011), given any $\boldsymbol{s}$ and $t$, we have

$$\mathbb{E}_{\boldsymbol{a}_0\sim\hat{\gamma}(\boldsymbol{a}_0|\boldsymbol{s})} \mathbb{E}_{\boldsymbol{a}_t\sim\nu_{t|0}(\boldsymbol{a}_t|\boldsymbol{a}_0,\boldsymbol{s})} \|s_\theta(\boldsymbol{a}_t; \boldsymbol{s}, t) - \nabla_{\boldsymbol{a}_t} \log \nu_{t|0}(\boldsymbol{a}_t|\boldsymbol{a}_0, \boldsymbol{s})\|_2^2$$

$$= \mathbb{E}_{\boldsymbol{a}_t\sim\nu_t(\boldsymbol{a}_t|\boldsymbol{s})} \|s_\theta(\boldsymbol{a}_t; \boldsymbol{s}, t) - \nabla_{\boldsymbol{a}_t} \log \nu_{t|0}(\boldsymbol{a}_t|\boldsymbol{s})\|_2^2 + C_{\boldsymbol{s},t}, \tag{40}$$

where $C_{s,t}$ is a constant to $\theta$ depending on $s$ and $t$. Substituting this result into the previous Eq. 39, we get

$$\mathcal{J}_{\text{HDSM}}(\theta) - \mathcal{J}_{\text{CSM}}(\theta) = \mathbb{E}_{s \sim \mu}\mathbb{E}_t w_t C_{s,t}. \tag{41}$$

Since the right-hand side is a constant to $\theta$, we have conclude that

$$\nabla_\theta \mathcal{J}_{\text{HDSM}}(\theta) = \nabla_\theta \mathcal{J}_{\text{CSM}}(\theta). \tag{42}$$

$\square$

# C. Details for Experiments

All experiments are conducted on an NVIDIA Tesla A100 80GB GPU, and all fine-tuning methods use the same pre-trained policy. In the pretraining, the observations and actions are normalized to $[0,1]$ using min/max statistics from the pre-training dataset. No history observation (pixel, proprioception, or ground-truth object states) is used. The deffusion policy is trained with learning rate $1e-4$ decayed to $1e-5$ with a cosine schedule, weight decay 1e-6 and 50 parallelized. For Franka-Kitchen and Robomimic tasks, epochs is 8000 and batch size is 128; for CALVIN tasks, epochs is 5000 and batch size is 512.

## C.1. Details and Hyper-parameters for OTPR

**Details for training $u_\omega, v_\omega$.** The architecture of both $u_\omega$ and $v_\omega$ is a two MLP. $\lambda$ is set to $1e-5$. The batch size is set 64. We employ the Adam algorithm to update the parameters with $1e-6$ learning rate.

**Details for training $s_\theta$.** We take the VP-SDE (Song et al., 2020b) as the forward SDE. In inference, we take the sampling method in DDIM (Song et al., 2020a) to perform the reverse SDE to generate action. The observations and actions are normalized to $[0,1]$ using min/max statistics from the pre-training dataset. For diffusion-based policies, we use MLP with two-layer residual connection similar to DPPO.

*Table 2.* Hyper-parameters for OTPR

| Parameter | Task | | | |
|---|---|---|---|---|
| | Franka-Ketichen | CALVIN | Robomimic-Can | Robomimic-Square |
| Buffer size | 1000000 | 250000 | 250000 | 250000 |
| Actor Learning Rate | 1.00E-05 | 1.00E-05 | 1.00E-05 | 1.00E-05 |
| Discount $\kappa$ | 0.99 | 0.99 | 0.999 | 0.999 |
| Optimizer | | Adam | | |
| $L$ | 8 | 8 | 8 | 8 |
| $T$ | 20 | 20 | 20 | 20 |
| $\tau$ | 0.7 | 0.7 | 0.7 | 0.7 |
| Actor Batch Size | 1024 | 1024 | 1024 | 1024 |
| Critic (Q and V) Hidden Layer Sizes | [512, 512, 512] | [512, 512, 512] | [256, 256, 256] | [256, 256, 256] |
| Critic (Q and V) Batch Size | 256 | 256 | 256 | 256 |

## C.2. Details and Hyper-parameters for Baselines

**DPPO** For the state-based tasks Robomimic and FrankaKitchen, we trained DPPO-MLP following the original paper's specifications, using an action chunking size of 4 for Robomimic and 8 for FrankaKitchen. For the pixel-based task CALVIN, we trained DPPO-ViT-MLP with an action chunking size of 4.

**IDQL** We employ the IDQL-Imp version of IDQL, wherein the Q-function, value function, and diffusion policy are refined through new experiences. For Robomimic tasks, we employ the same network architecture as OTPR, while the original IDQL architectures are preserved for Franka-Kitchen and CALVIN. For the IQL $\tau$ expectile, we set it to 0.7 for each task.

**DQL** We set the weighting coefficient to $0.5$ for Robomimic, $0.005$ for Franka-Kitchen and $0.01$ for CALVIN.

**IBRL** We adhere to the original implementations' hyperparameters, with wider (1024) MLP layers and dropout during pre-training.

**Cal-QL** We set the mixing ratio to 0.25 for Franka-Kitchen and 0.5 for CALVIN and Robomimic.

# D. Additional Experimental Results

## D.1. Pixel-based Tasks in Robomimic

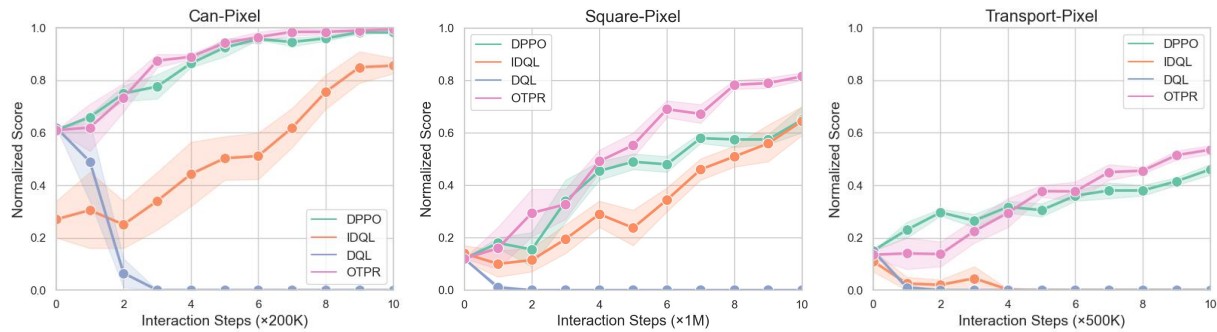

*Figure 5.* **Learning curves of online fine-tuning with various methods on pixel-based Robomimic Tasks.** Observe that OTPR largely always dominates or attains similar performance to the next best method. Other methods for fine-tuning diffusion policies (IDQL, DQL, DPPO) are a bit unstable, and perform substantially worse.

## D.2. LIBERO-Long Tasks

*Table 3.* Success rate of online fine-tuning on 10 task in Libero-Long.

| Libero-Long | Task 1 | Task 2 | Task 3 | Task 4 | Task 5 | Task 6 | Task 7 | Task 8 | Task 9 | Task 10 |
|---|---|---|---|---|---|---|---|---|---|---|
| DP | 0.65 | 0.5 | 0.85 | 0.7 | 0.6 | 0.8 | 0.6 | 0.65 | 0.1 | 0.7 |
| OTPR | 0.85 | 0.65 | 0.85 | 0.9 | 0.8 | 0.9 | 0.7 | 0.9 | 0.35 | 0.85 |

## D.3. Qualitative Toy Experiments

**Compatibility Function Validation**

Objective: Validate the accuracy of the compatibility function to verify whether Algorithm 2 can effectively learn the dual term.

Setup: We conducted experiments in a 2D space, where Algorithm 2 was applied to a random dataset consisting of a Gaussian source distribution and a multi-modal target distribution (8 Gaussian), with the Euclidean distance serving as the cost function.

Fig. 6 left: We visualize the source distribution as colored level sets and the target distribution as randomly sampled points.

Result 1 (Fig. 6 middle): We constructed 200 sample points from the source distribution and 2000 samples from the target distribution, then paired them using the compatibility function $H$. The compatibility function H successfully matches source samples (yellow) to target samples (green) with low cost, confirming its effectiveness.

Result 2 (Fig. 6 right): Generated samples obtained by sampling from the source distribution. We learn an optimal map as a neural network by approximating the barycentric projection of the OT plan from Algorithm 2. Generated samples closely match the target distribution.

**OT-Guided Diffusion Policy Verification**

Objective: Assess diffusion policy's ability to recover target distributions under OT guidance.

Setup: We leverage 2 synthetic 2D datasets (8Gaussian and swissroll) to further verify the effectiveness of OT guided diffusion policy. Each dataset contains data points paired with specific Q values (Figure Fig. 7 left).

Results (Fig. 7 right): The ultimate samples generated by the diffusion model, which closely match the ground-truth target distribution.

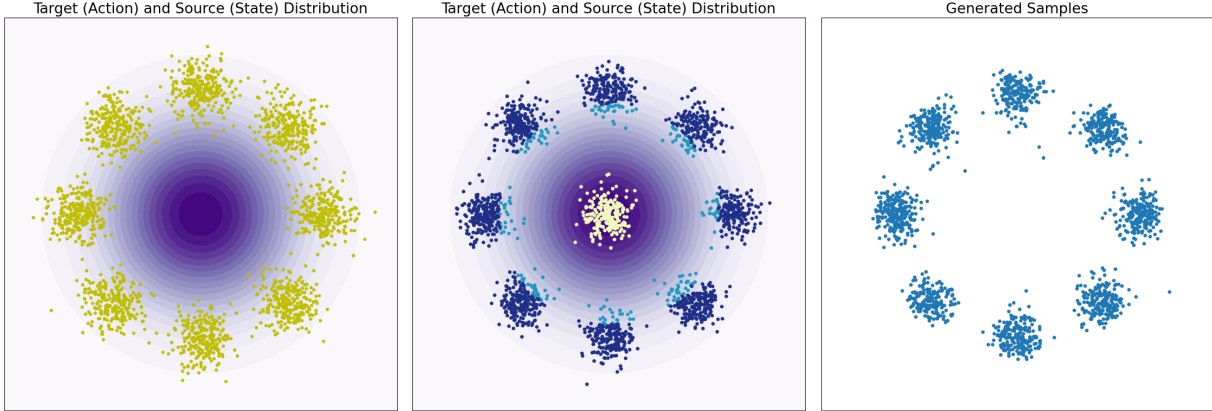

*Figure 6.* **Left**: A continuous Gaussian source distribution (colored level sets) and a multi-modal target samples from 8 Gaussian distribution (yellow). **Middle**: Samples from source distribution (yellow), paired samples (green) and unpired samples (blue) from target distribution. **Right**: Generated samples by approximating the barycentric projection of the OT plan.

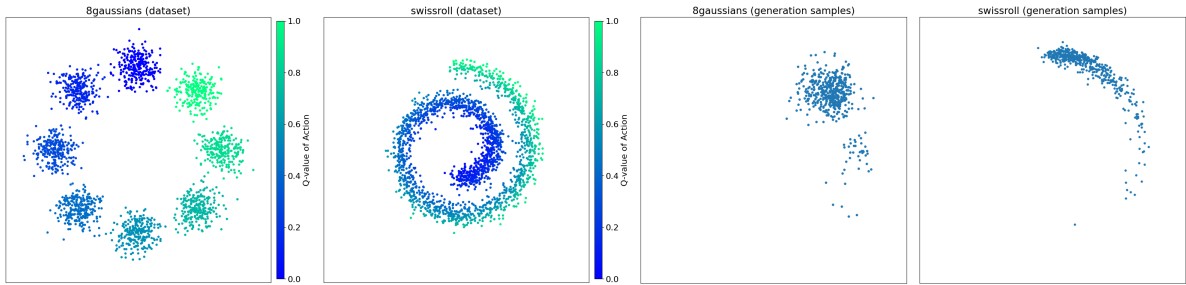

*Figure 7.* Illustration of the 8gaussians and swissroll dataset (left) and the generation result of diffusion model (right)

