# OpenReview forum: "Score-Based Diffusion Policy Compatible with Reinforcement Learning via Optimal Transport"
_ICML.cc/2025/Conference — ICML 2025 poster_

### Official Review · Reviewer_an3h · 2025-03-08

**Overall Recommendation:** 3

**Summary:**

The method proposed in this paper is quite complex, but I will do my best to summarize it:
* The authors attempt to integrate diffusion policies (which are typically learned via imitation on expert data) with online environment interactions.
* To this end, the authors make the following key observation (Prop 4.1): minimizing the optimal transport problem from expert state distribution -> expert action distribution, with the cost being the negative of the expert Q function, yields a policy of equivalent performance to the expert.
* So our algorithm is roughly repeats the following steps:

a) Learn the optimal transport problem solution (as dual functions, although we can recover the primal), given the current learned Q-function and data from the replay buffer.

b) For each inference step, sample a bunch of state-conditioned actions from the diffusion policy. Weight the probabilities state-action pairs by the coupling function H(s, a) from the OT problem solution; intuitively, we are doing weighted sampling based on how "likely" it is that an optimal policy following the current Q-function would take each action. Select an action according to the reweighted probabilities and execute in the environment.

c) After doing a rollout, update the Q-function based on the observed rewards.

d) Update the diffusion policy using equation (10); this is similar to advantage-weighted regression, where we are upweighting the state-action pairs where H(s, a) is high.

* Basically, if my understanding is correct, we solve the OT problem transform a Q function into this coupling object H(s, a), which tells us how "good" a state-action pair is. We then use H to both weight (s,a) pairs in training the diffusion policy and guide sampling at inference time.
* None of the above strictly requires expert demonstrations. But if we do have expert demonstrations, these come in the OT solving step via a masking function which gives keypoints to guide the OT solution.
* The authors show convincing performance of this method across a range of problems.

**Claims And Evidence:**

Claim: the OTPR policy outperforms competing baselines.
- Evidence: strong. Across all experiments in Figure 2, OTPR is either clearly best or tied. It is also clearly the most consistent--no other baseline performs close to OTPR for all scenarios. The performance improvement compared to other demo-augmented RL algorithms in Talbe 1 is quite substantial.

Claim: the guidance from coupling function H (for training and sampling at inference) is better than that from the Q function or the advantage function.
- Evidence: moderate to strong. The authors compare these for a single robomimic-can task (Figure 3 left), where the difference is substantial, but I'd like to see this for other environments as well.

Claim: OTPR without the expert-demonstration masking "exhibits instability and reduced efficiency" (420).
- Evidence: weak to moderate. The distinction between the masked and unmasked versions of OTPR in Figure 3 right are marginal, and again only for a single environment. I don't see what suggests that the unmasked version "exhibits instability".

**Essential References Not Discussed:**

N/A

**Experimental Designs Or Analyses:**

The experiments are reasonable. I like that the hyperparameters for the OTPR are consistent across all the tasks in Table 2 (Appendix C.1), meaning that the method doesn't require extensive hyperparameter tuning.

**Methods And Evaluation Criteria:**

Yes, and they are thorough.

**Other Comments Or Suggestions:**

N/A

**Other Strengths And Weaknesses:**

My main reservation is that the paper is pretty hard to understand. There's a lot going on, the math exposition is pretty confusing, and I think there are quite a few typos. Here are some of my confusions:
1. In line 166 RHS, the authors say that the policy "moves mass from the state distribution $\mu(s)$ to the distribution of actions $\nu(a)$." What are these state and action distributions? Are they the stationary distributions under the expert policy? Effectively, the marginals distributions of the (s,a) pairs from the expert dataset?
2. In section 4.2, the authors denote the "condition data" of an action as $s_{cond}(a)$. This makes very little sense to me, as the same action can be taken in multiple different states, so there isn't one unique state that produces a particular action.
3. I think in the RHS of 13 the i index should appear in the numberator, instead of k?
4. The authors say that the input to Algorithm 1 is an "initailized Q-network"-- what does this mean? Is it pretrained?
5. In (22), should it read $\pi(s) \in \text{Supp}(\nu)$?
6. In the proof of Prop B.1, the relevant result from Kakade & Langford should be reproduced to make everything self-contained.
7. When introducing the Regularized OT Dual, it should be explicitly stated what kind of objects u and v are. I think it would be also nice if the derivation of the dual could be provided in the appendix because it's not obvious to me how that happened.

**Questions For Authors:**

1. What you show is that H(s, a) is effectively just a "better" version of the Q function (Figure 3 left) for guiding and training diffusion policies. It seems that H(s, a) comes from three main ingredients: the Q function, the marginal distribution over states, and the marginal distribution over actions. What, intuitively, do these last two ingredients "add in" to make H better than Q? It seems in algorithm 2 that you are just learning H from the data in the replay buffer, which is collected from a suboptimal policy (the one currently being trained). Why would that help?
2. In figure 4, why isn't optimal transport assigning a unique action for every state? Am I right that this is the relaxed scheme (eq (2)) instead of the Monge formulation (eq (1))?

**Relation To Broader Scientific Literature:**

This paper is addressing a really important topic in robotics right now: how can we improve diffusion policies with online data? This is a challenging problem for diffusion policies in particular. Typically, in online RL you need to take the gradient of the action likelihood with respect to the policy parameters; but for diffusion policies, there is no closed-form action likelihood as actions produced via an iterative sampling scheme.

A natural idea is to weight the regression targets in the diffusion loss by the Q function or the advantage. I'm mentally understanding the submission as proposing a better version of this using an OT coupling function H(s, a) instead. The experiments are pretty compelling, even though I don't fully understand the derivation.

**Theoretical Claims:**

I did have a look at the proof of Proposition B.1 and it seems reasonable, although I have some confusions about the problem setup which means I wasn't able to confidently check it (see below).

---

> ### Author Rebuttal · Authors · 2025-04-01
>
> Dear Reviewer an3h,
>
> Thank you for your detailed review and constructive feedback on our submission. Below, we address your main concerns and questions.
>
> ## C1: Claims And Evidence:
>
> **R1:** We appreciate the feedback regarding evidence granularity. To substantiate our claims, we have conducted additional experiments on the Robomimic-Square environment (see Figure 4 in  https://anonymous.4open.science/r/OTPR_Supplementary-5F55/README.md). The observed "instability" manifests as OTPR-U producing worse fine-tuning outcomes more frequently across multiple trials, which is visually reflected in the increased variance of success rates. We agree the original terminology could be misinterpreted and will revise "instability" to "statistically significant variance increase".
>
> ## C2: Weaknesses
>
> **R2:** We acknowledge that the confusing math exposition are weaknesses. We'll work to improve the clarity of the mathematical derivations and the overall exposition. Below, we address your main confusion:
>
> **1.** Here, we assume the existence of an stationary optimal behavior policy, $\pi^\beta$ in standard MDP. In this context, the "state distribution" is formally defined as the idealized stationary distribution induced by the expert policy. This theoretical abstraction serves to formulate the imitation learning problem as a Monge optimal transport (OT) problem between distributions. We will clarify it.
>
> However, we explicitly recognize that: The true stationary  state/action distribution is intractable to compute directly in practice, especially for complex environments. Real-world expert datasets only provide finite samples from this distribution. So we introduce a neural network-based approximation in 4.3 to estimate the underlying OT plan on expert dataset.
>
> **2.** The reviewer is correct; we did not assume a one-to-one mapping between state and action here. If the term "condition data" has caused confusion, we are willing to revise it to "all conditional states."
>
> **3.** Thank you for the reminder. Indeed, there was an error on our part, and we will correct it.
>
> **4.** The "initialized Q-network" refers to a newly instantiated Q-network within the deep RL that has not undergone any training iterations.  To enhance algorithmic transparency, we propose adding an explicit initialization declaration step in the pseudocode.
>
> **5.** The reviewer is correct. Thank you for the reminder and we will correct it.
>
> **6.** We directly employed **Lemma 6.1** from (Kakade & Langford, 2002). As suggested by the reviewers,
> we will explicitly clarify the equivalence between the advantage function $A$ and $R$ to enhance the readability of the derivation.
>
> **7.** Thanks to the reviewer for the suggestion. In order to facilitate a clear understanding of Regularized OT Dual for the reader, we will additionally provide an introduction to Kantorovich duality and provide a proof in the appendix, the core idea is rewrite the constrained infimum problem as an **inf** **sup** **problem**, and exchange the two operations by formally applying a minimax principle,i.e. replacing an “inf sup” by a “sup inf”.
>
> ## C3: Question 1 For Authors
> *H(s, a) as a Better Version of the Q Function?*
>
> **R3:** First, we would like to clarify the reviewer's understanding. As described in Equation 15, **the compatibility function $H$ incorporates two estimated dual variables $u$ and $v$, rather than the distributions $\mu$ and $\nu$**. We sincerely apologize for the unintended ambiguity caused by the visual similarity between English and Greek letters in our notation. In the revised manuscript, we will adopt more distinctive notation to better differentiate these variables.
>
> Secondly, imitation learning aims to acquire a deterministic state-action mapping from expert demonstrations, while reinforcement learning focuses on learning a Q-function to evaluate action sampling, even for suboptimal actions. Our approach bridges these two objectives by introducing a compatibility function $H$ that establishes a **soft** state-action coupling relationship from a data distribution perspective. Specifically, the proposed compatibility function offers two advantages: (1) For clearly advantageous state-action pairs (e.g., those from expert demonstrations), $H$ provides precise guidance; (2) For novel state-action pairs that may emerge during RL exploration (particularly those absent from the training data), where Q-value estimates could be unreliable, the potential functions derived from the optimal transport plan serve as a corrective mechanism to adjust these estimates.
>
> ## C4:  Question 2 For Authors
> *In figure 4, why isn't optimal transport assigning a unique action for every state? Am I right that this is the relaxed scheme (eq (2)) instead of the Monge formulation (eq (1))?*
>
> **R4:** The reviewer is correct. As indicated in the title of Figure 4, we indeed visualize the Optimal Transport Plan in this figure.

---

> > ### Comment · Reviewer_an3h · 2025-04-03
> >
> > Thank you to the authors for their thorough response. A few comments on my remaining concerns below:
> >
> > *The reviewer is correct; we did not assume a one-to-one mapping between state and action here. If the term "condition data" has caused confusion, we are willing to revise it to "all conditional states."* The formulation of Proposition 4.2 doesn't make sense if $s_{cond}$ is now a set-valued map. What does $\mathcal{C}(s, a)$ become? Can the authors describe in detail where $s_{cond}$ comes from in practice?
> >
> > *For novel state-action pairs that may emerge during RL exploration (particularly those absent from the training data), where Q-value estimates could be unreliable, the potential functions derived from the optimal transport plan serve as a corrective mechanism to adjust these estimates.* Can you elaborate on this? Why are the potential functions not also unreliable, since you're using the Q function as the optimal transport loss? How do the potential functions act "as a corrective mechanism"?

---

> > > ### Author Response · Authors · 2025-04-08
> > >
> > > Dear Reviewer,
> > >
> > > Thank you very much for your insightful comments and constructive feedback. We regret that our previous rebuttal may not have fully addressed your concerns due to word limitations. Please find our detailed response to each of your points:
> > >
> > > ## C1. What does $\mathcal{C}(s,a)$ become? Can the authors describe in detail where $s_{cond}(a)$ comes from in practice.
> > >
> > > **R1:**
> > > 1. The reviewer’s observation is entirely valid. When
> > > $s_{cond}(a)$ is a set-valued map (i.e., an action
> > > $a$ may correspond to multiple states
> > > $s$), the original definition of the Dirac delta function $\delta(s-s_{cond}(a))$ introduces a mathematical inconsistency, as $s_{cond}(a)$ becomes a set rather than a single state. To resolve this, the **Generalized Dirac Measure** $\delta_{s_{cond}(a)}(s)$ can be defined as an integral measure over sets, i.e., $\delta(s) = 0$, if $s \notin s_{cond}(a)$.
> > > 2. Our goal is to achieve smooth RL fine-tuning for diffusion policy (DP) from imitation learning (IL). So we introduce Proposition 4.2 to reformulate the IL objective $J_{DSM}$ of DP into $J_{CDSM}$, which establishs a mapping from states to actions through data-driven learning. Thus, the context here is the setting of imitation learning, and $s_{cond}(a)$ originates from the paired expert data of $(s, a)$. The actual code implements this by constructing a hash map.
> > >
> > >
> > > We greatly appreciate the reviewer's comments. We will make the necessary corrections and add relevant explanations in the manuscript. This will make our theory more standardized and clear.
> > >
> > > ## C2. For novel state-action pairs ... Can you elaborate on this? Why are the potential functions not also unreliable, since you're using the Q function as the optimal transport loss? How do the potential functions act "as a corrective mechanism"?
> > >
> > > **R2:**
> > > Thank you for the insightful question. Our method shares conceptual similarities with offline RL approaches like **Weighted Regression** (e.g. AWR and RWR)and **Selection from Behavior Candidates** (e.g. SfBC and IDQL). Both our method and prior works (e.g., IDQL and SfBC) involve: **(1) Using Q-learning to assign scores to state-action pairs from the behavior policy.**
> > > **(2) Training a diffusion policy via forward KL minimization.**
> > >
> > > Obviously, a naive approach would **directly resample actions using Q-values** as weights (as in SfBC), but this risks over-reliance on Q-values for OOD pairs, where they may be unreliable or falsely high.
> > >
> > > ***Corrective Role of Potential Functions:***
> > >
> > > Instead of relying solely on Q-values, we use the optimal transport (OT) plan to derive weights. Specifically:
> > >
> > > The OT plan’s dual potentials (estimated from dataset and replay buffer) decouple the dependency on Q(s,a) for novel pairs by separately reweighting states (s) and actions (a) based on their marginal distributions.
> > >
> > > For in-distribution (s,a): Q-values are relatively accurate, so the OT-derived potentials (and the resulting composite cost H(s,a)) align with Q-learning.
> > >
> > > For OOD (s,a): The potentials act as a conservative regularizer by leveraging the **global** structure of the dataset (via state/action marginals) rather than trusting local Q-extrapolations.
> > >
> > > ***Why Potentials Are More Reliable Than Q-Learning Alone:***
> > >
> > > The potentials are not trained to maximize returns (unlike Q-functions) but to approximate the data distribution’s geometry (via OT’s marginal constraints).
> > >
> > > While the OT loss uses Q-values as a cost, the potentials are smoothed over the dataset—avoiding overfitting to spurious Q-peaks. This is analogous to how OT-based imputation handles noisy inputs by enforcing mass conservation.
> > >
> > > The MASK mechanism further ensures that expert data retain their original pairs.
> > >
> > > In essence, the OT plan provides a conservative reweighting that balances Q’s local accuracy with global distributional fidelity.

---

### Official Review · Reviewer_ZmNb · 2025-03-12

**Overall Recommendation:** 4

**Summary:**

This paper introduces OTPR, a novel method that integrates optimal transport theory with diffusion policies to enhance the robustness and performance of imitation learning models through online interactions with the environment . The core algorithmic idea involves leveraging the Q-function as a transport cost and viewing the policy as an optimal transport map to establish a connection between optimal transport and RL . OTPR also introduces masked optimal transport to guide state-action matching using expert data as keypoints and a compatibility-based resampling strategy to improve training stability. The paper's main findings from experiments on three simulation tasks demonstrate that OTPR consistently matches or outperforms existing state-of-the-art methods, especially in complex and sparse-reward scenarios, highlighting its effectiveness in combining imitation learning and reinforcement learning for versatile and reliable policy learning.

## update after rebuttal
I confirm my score. Authors addressed comments and added clarity and results to the original submission.

**Claims And Evidence:**

The paper’s claims that OTPR (1) integrates optimal transport with diffusion policies to stabilize fine-tuning, (2) achieves notable performance gains over baseline methods, and (3) remains robust in sparse-reward environments—are generally well-supported. Multiple experiments on robotic tasks with varying difficulty, along with comparisons to several recent diffusion-based and demo-augmented RL baselines, underscore OTPR’s improvements. The authors also include ablations (e.g., masked vs. unmasked OT) to illustrate how each component contributes to the final performance.

**Essential References Not Discussed:**

No

**Experimental Designs Or Analyses:**

The paper’s ablation studies that remove “masked OT” or the compatibility-based resampling strategy appear valid for isolating each component’s effect on performance. The comparisons to both diffusion-based and non-diffusion-based RL methods also lend credibility to the authors’ main performance claims. However, finer details like runtime comparisons are treated less thoroughly, so it’s hard to assess how sensitive or resource-intensive the method might be in broader settings. Overall, the key experiments are logically consistent, and no major flaws are evident in their design or analysis.

**Methods And Evaluation Criteria:**

The paper takes Robomimic, Franka-kitchen and CALVIN as the evaluation benchmark, which are suitable for assessing multi-step tasks under distribution shifts, making the evaluation well-matched to the paper’s goals.

**Other Comments Or Suggestions:**

Typos in Section 3.2: “Reinfrocement Learning” should be “Reinforcement Learning”.

**Other Strengths And Weaknesses:**

**Strengths:**

- OTPR is the first method that combines diffusion based policy, RL and OT.
- Propose Mask Optimal Transport in RL finetuning.

**Weakness:**

- Still need to test in Real-world environment like DPPO. Because performance achieved in a simulated environment may not directly translate to equivalent real-world performance.

**Questions For Authors:**

How long is fine-tuning the diffusion policy with OTPR?

**Relation To Broader Scientific Literature:**

OTPR unifies insights from generative modeling, reinforcement learning, and OT to tackle distribution mismatch more effectively.

**Theoretical Claims:**

There are no apparent flaws in the theoretical arguments as presented.

---

> ### Author Rebuttal · Authors · 2025-04-01
>
> Dear Reviewer ZmNb,
>
> Thank you for your positive review and valuable feedback on our submission. Your comments have provided us with clear directions for improvement, and we are committed to addressing them in the revised version of our paper. Below, we address your main concerns and questions.
>
> ## C1: Questions For Runtime
>
> **R1:** We thank the reviewers for their suggestion. Since all experiments were conducted under the same computational resource configuration, we directly provide the wall-clock time statistics for comparison in the Table of Supplementary Section 3 (https://anonymous.4open.science/r/OTPR_Supplementary-5F55/README.md).
>
> Our algorithm incurs approximately **8%–11%** more runtime compared to other baselines. This additional overhead stems from learning the dual term, but it is negligible considering the performance improvements achieved.
>
> ## C2: Weakness (Real World Experiments)
>
> **R2:** We sincerely appreciate the reviewer’s valuable feedback regarding the importance of real-world validation. We fully acknowledge that performance in simulated environments may not directly translate to real-world scenarios. However, due to the time constraints of the rebuttal period and the current resource limitations, we are unable to conduct real-world experiments at this stage. As demonstrated in DPPO deployments, policies trained in high-fidelity simulations can achieve zero-shot transfer to physical hardware without real-data fine-tuning. So we have justifiable confidence that our approach can achieve comparable or better zero-shot transfer performance under identical experimental settings. We are currently actively developing a simulation-to-real evaluation platform and commit to publishing empirical validation results by the Camera-Ready deadline.
>
>
> ## C3: Typos and Terminology
>
> **R3:** Thank you for catching the typo in Section 3.2. We will correct "Reinfrocement Learning" to "Reinforcement Learning" in the final version of the paper. We appreciate your attention to detail.
>
>
> Thank you again for your time and insightful comments.

---

> > ### Comment · Reviewer_ZmNb · 2025-04-03
> >
> > Thank you for the clarifications regarding runtime and future plans for real-world experiments.
> >
> > That said, I remain concerned about the limited experimental scope. While I understand real-robot evaluations may be constrained, the current benchmarks (kitchen, robomimic, CALVIN) are relatively standard and do not fully test the claimed benefits of OTPR in sparse-reward, high-dimensional, or long-horizon settings. There exist more challenging and diverse simulation tasks, such as Robomimic’s transport-pixel/state and furniture benchmarks compared with DPPO, that could have offered a more convincing demonstration.
> >
> > Without such experiments, it is difficult to fully assess the robustness and generality of the proposed method. I am currently keeping my score at 4, but I would strongly encourage the authors to include more diverse and challenging tasks in the final version.
> >
> > Otherwise, I may reconsider my recommendation.

---

> > > ### Author Response · Authors · 2025-04-09
> > >
> > > Dear Reviewer,
> > >
> > > Thank you for your constructive feedback and for understanding the constraints on real-robot evaluations. We appreciate your suggestions to strengthen our work.
> > >
> > > In response to your concerns, we have conducted additional experiments to further demonstrate the robustness of OTPR:
> > >
> > > 1. We have already evaluated OTPR on the **pixel-based Robomimic task** during the rebuttal period (Figure 2 (https://anonymous.4open.science/r/OTPR_Supplementary-5F55/README.md).  The experimental results clearly demonstrate that our method either significantly outperforms or achieves comparable performance to the next best baseline approach.
> > >
> > > 2. We further evaluated OTPR’s fine-tuning capabilities on **LIBERO-Long**, which underscores its effectiveness in long-horizon tasks. The success rate is shown on the Table of Supplementary Section
> > >  5 (https://anonymous.4open.science/r/OTPR_Supplementary-5F55/README.md). Due to time constraints, detailed results (including comparisons of different finetuning methods and more tasks) will be progressively updated on the anonymized GitHub repository. Full analyses and real-robot experiment will also be included in the camera-ready paper.
> > >
> > > We sincerely appreciate your guidance and hope these additions address your concerns. Thank you again for your support.

---

### Official Review · Reviewer_iuX1 · 2025-03-14

**Overall Recommendation:** 3

**Summary:**

The paper proposes OTPR that leverages optimal transport for fine-tuning diffusion policy in RL. Q function is treated as the transport cost and the policy is considered the transport map. Masked OT with resampling is also applied to improve training stability. Experiment results show generally improved performance compared to other diffusion RL fine-tuning methods and demo-augmented RL methods.

**Claims And Evidence:**

The claim of improved performance is supported by results on the Franka-Kitchen, CALVIN, and Robomimic tasks, which are manipulation tasks and generally more challenging than dense-reward mujoco locomotion tasks.

However, the paper lacks qualitative discussion on its effectiveness. I would vote for strong accept if there are qualitative demonstrations of improved performance from leveraging optimal transport, e.g., in a carefully designed toy problem.

**Essential References Not Discussed:**

A missing reference is Psenka et al., Learning a diffusion model policy from rewards via q-score matching, which also considers online diffusion-based RL with Q function.

**Experimental Designs Or Analyses:**

I do not spot any particular issue with the experimental design.

**Methods And Evaluation Criteria:**

I appreciate the author considering more challenging RL benchmarks including the vision-based CALVIN.

**Other Comments Or Suggestions:**

I suggest putting more experimental details, such as how the diffusion policies are pre-trained, in the beginning of experiment section.

**Other Strengths And Weaknesses:**

The paper is well-written and figures (especially ones in the experiment section) are nicely done.

**Questions For Authors:**

Did you try pixel-based robomimic tasks? Would also be interesting to try Transport from robomimic, which is significantly more challenging than the tasks considered.

**Relation To Broader Scientific Literature:**

Diffusion RL fine-tuning is a very important area that requires active research as we have seen many successes in training diffusion policy with behavior cloning, and RL fine-tuning is critical to further improve the robustness. This paper bridges the gap between Ren et al. in online diffusion-based RL and many existing offline diffusion-based RL methods.

**Theoretical Claims:**

I skimmed over the proofs in Appendix B and did not see any egregious error.

---

> ### Author Rebuttal · Authors · 2025-04-01
>
> Dear Reviewer iuX1,
>
> Thank you for your positive review and constructive feedback on our submission. We appreciate your recognition of our work and are glad to hear that you found our approach and experimental results valuable. Below, we address your specific comments and questions.
>
> ## C1: Qualitative Experiments in Claims And Evidence
>
> **R1:** As suggested, we designed two illustrative 2D toy experiments to visually validate our method’s core components. Full visualizations are available in [Supplementary Section 1] (https://anonymous.4open.science/r/OTPR_Supplementary-5F55/README.md).
>
> ### 1. Compatibility Function Validation
>
> **Objective:** Validate the accuracy of the compatibility function to verify whether Algorithm 2 can effectively learn the dual term.
>
> **Setup:** We conducted experiments in a 2D space, where Algorithm 2 was applied to a random dataset consisting of a Gaussian source distribution and a multi-modal target distribution (8Gaussian), with the Euclidean distance serving as the cost function.
>
> **Figure 1.1 left:** We visualize the source distribution as colored level sets and the target distribution as randomly sampled points.
>
> **Result 1 (Figure 1.1 middle):** We constructed 200 sample points from the source distribution and 2000 samples from the target distribution, then paired them using the compatibility function H. The compatibility function H successfully **matches source samples (yellow) to target samples (green) with low cost**, confirming its effectiveness.
>
> **Result 2 (Figure 1.1 right):**
> Generated samples obtained by sampling from the source distribution. We learn an optimal map as a neural network by approximating the barycentric projection [1] of the OT plan from Algorithm 2. Generated samples closely match the target distribution.
>
> ### 2. OT-Guided Diffusion Policy Verification
> **Objective:** Assess diffusion policy’s ability to recover target distributions under OT guidance.
>
> **Setup:** We leverage 2 synthetic 2D datasets (8Gaussian and swissroll) used in [2] to further verify the effectiveness of OT guided diffusion policy. Each dataset contains data points paired with specific Q values (**Figure 1.2 left**).
>
> **Results (Figure 1.2 right):** The ultimate samples generated by the diffusion model, which closely match the ground-truth target distribution.
>
> [1] Seguy, Vivien, et al. "Large-Scale Optimal Transport and Mapping Estimation." ICLR 2018-International Conference on Learning Representations. 2018.
>
> [2] Lu, Cheng, et al. "Contrastive energy prediction for exact energy-guided diffusion sampling in offline reinforcement learning." International Conference on Machine Learning. PMLR, 2023.
>
>
>
>
> ## C2. Experimental Details
>
> **R2:**  Thanks for the reviewer’s suggestion, we will supplement the more relevant details of the pre-trained diffusion policy in Section 6 or Appendix C if the page number is limited.
>
> *In the pretraining, the observations and actions are normalized to [0, 1] using min/max statistics from the pre-training dataset. No history observation (pixel, proprioception, or ground-truth object states) is used. The deffusion policy is trained with learning rate 1e-4 decayed to 1e-5 with a cosine schedule, weight decay 1e-6 and 50 parallelized. For Franka-Kitchen and Robomimic tasks, epochs is 8000 and batch size is 128; for CALVIN tasks, epochs is 5000 and batch size is 512.*
>
>
>
> ## C3. Missing Reference
>
> **R3:** We appreciate your suggestion to include **QSM** in our manuscript, which is closely related to our approach and offers valuable insights into the field. We will reference this paper appropriately in the Related Work section.
>
>
>
> ## C4. Additional Robomimic Task
>
> **R4:** We sincerely appreciate this constructive feedback. To address the request, we have conducted additional experiments on pixel-based tasks in Robomimic, including the Transport task highlighted by the reviewers. We used ResNet as the visual encoder, similar to the setup in CALVIN. The learning curves and comparisons with baseline methods can be found at (https://anonymous.4open.science/r/OTPR_Supplementary-5F55/README.md).
>
> As shown in Figure 2 from the linked results, our method still dominates or attains similar performance to the next best method. The results align with our original pixel-based CALVIN experiments, further validating our framework’s capability to handle visual inputs while preserving offline-to-online finetuning strengths.

---

### Official Review · Reviewer_L6dM · 2025-03-17

**Overall Recommendation:** 3

**Summary:**

This paper proposes to reformulate offline-to-online diffusion policy training with optimal transport. It views policy as a transport from the state distribution to action distribution, using the (negative) Q-function as a a transport cost and treating the policy as an optimal transport map. The authors show that the score matching objective of the diffusion policy training can be augmented with a weighting function that constitutes the joint distribution of state and action pairs. They further show that this weighting function can be relaxed with a compatibility function that involves the Q function as well as some dual variables $u_w(\mathbf{s})$ and $v_w(\mathbf{a})$ derived from the dual form of regularized optimal transport. As this relaxed weighting function gives zero weight to state-action pairs where the state is in expert demonstrations while the action is not, they coin their objective Masked Optimal Transport. The authors also did an analysis to illustrate that the the proposed training algorithm is optimizing an upper bound of the distance between the diffusion policy and the optimal transport plan. Their experiments demonstrate clear offline-to-online improvement.

### After rebuttal ###
The authors addressed my major concerns. I updated my rating accordingly.

**Claims And Evidence:**

The major claim is that the OT perspectives help bridge diffusion policy with RL, which seems to be only weakly supported by showing the OT is using the (negative) Q function as a cost function. However, this connection appears to be fairly superficial since which Q learning method is used and what policy induces its Q value is not discussed.

**Essential References Not Discussed:**

N/A

**Experimental Designs Or Analyses:**

The authors conducted experiments in online funetuning on 6 RL tasks, which seems not very sufficient. Fig 2 shows the proposed method has significant effect in most of them.

They also compare with methods that use both offline data that are not necessarily optimal and expert data in offline training and online finetuning. The proposed method appears to be the only effective one in offline training, and performance is boosted after online finetuning.

**Methods And Evaluation Criteria:**

Viewing policy learning as an optimal transport from state distribution to action distribution makes sense conceptually. However, how to define the transportation cost is tricky. The proposed method has an implicit assumption that the key states, i.e. states covered by expert demonstrations, can only be paired with the associated actions in demonstrations. This appears to be a restriction to generalization.

The evaluation criteria, i.e. the learning curves and the final performance, are standard in the literature.

**Other Comments Or Suggestions:**

The notations appear to be very unorganized. For example, $\mathbf{a}$ and $a$ are used interchangeably in Section 3.2.

There is a typo in Eq 11.

**Other Strengths And Weaknesses:**

\+ The ablation of different compatibility method clearly illustrate the effectiveness of the proposed H function.

\- The formulation appears to be complicated. I appreciate the authors' effort in introducing the perspectives of optimal transport to diffusion policy. But the exposition of the paper involves non-intuitive formal notions, making it less accessible. The proposed algorithm also look complicated. At each iteration, the dual variables, which are MLPs need to be "optimized" as the first step of learning.
\- The proposed masked OT does not introduce significant gains according to FIg3.

**Questions For Authors:**

The mask scheme that only allows states in the expert demonstration to be matched with the actions appears to be pretty restrictive. Does that mean the model won't discover any actions that are equivalently good to actions that appear in the demonstration?

How is the offline learning in Table 1 performed? Why does the proposed method performs well in offline learning while most of the baseline methods failed?

**Relation To Broader Scientific Literature:**

The problem of offline-to-online learning is crucial for robotics. This paper is an attempt in this direction.

**Theoretical Claims:**

The proofs make sense to me at a high level. But I didn't check the details.

---

> ### Author Rebuttal · Authors · 2025-04-01
>
> Dear Reviewer,
>
> Thanks for your thorough review and constructive feedback on our submission.
>
> ## C1. Claims:
> *The connection between OT and RL appears superficial, and the role of Q-learning methods requires clarification.*
>
> **R1:** While the cliam has already garnered recognition from other reviewers (e.g. *OTPR unifies insights from generative modeling, reinforcement learning...* by Reviewer ZmNb), we provide these focused responses to your concerns:
>
> **Theoretical Equivalence:**
> To support this claim, we have provided proof in the Appendix B.1 demonstrating the equivalence between the OT plan and the optimal policy. This theoretical result forms a solid foundation for our approach and shows that the connection is not merely superficial.
>
> **The Relevance of Q as cost:** The use of the Q-function as a transport cost has built on prior static offline RL works (e.g., [1]). Our work uniquely integrates this idea with diffusion policy to enable smooth offline-to-online finetuning.
>
> **Q-Learning Compatibility:** OTPR is an algorithm-agnostic framework incorporating any existing Q learning algorithms.
>
> [1] Asadulaev, Arip, et al. "Rethinking Optimal Transport in Offline Reinforcement Learning." Advances in Neural Information Processing Systems (2024)
>
> ## C2. Generalization:
> *The mask appears to be a restriction to generalization.*
>
> **R2:** We appreciate the reviewer’s attention to generalization capabilities. We believe the authors have understood that our algorithm consists of two parts: (1) estimating the OT plan to provide a compatibility function H(s,a), and (2) using H to guide the  diffusion policy optimization.
>
> The mask is introduced during (1) and designed to fully leverage the paired state-action data from expert demonstrations to improve the accuracy of $H$. The effectiveness of this key-point guidance approach has been validated in some domain adaptation studies.  Crucially, the mask does not constrain the policy’s action space. The mask focuses only on actions that are known to be effective in specific seen states, as demonstrated by expert data, and does not directly influence the learning of the policy and inference. During online fine-tuning, the policy can freely explore novel actions in both seen and unseen states.
>
> ## C3. Additional Experiments:
>
> **R3:**  We additionally added two 2D qualitative toy experiments and three pixel-based Robomimic tasks results in https://anonymous.4open.science/r/OTPR_Supplementary-5F55/README.md. See response to reviewer iuX1 for a detailed description.
>
> ## C4. Ablation Results of Masked OT:
>
> **R4:** The masked OT mechanism is designed to refine the estimation of the OT plan by leveraging expert demonstrations as high-confidence priors. Its contributions are twofold: (1) reducing the occurrence of poor solutions, and (2) improving overall stability across multiple fine-tuning evaluations. As shown in Figure 3, these improvements are captured by the consistent trend in mean performance and the reduction in variance (tighter confidence intervals). We also supplement additional ablation experiments (Figure 4 in https://anonymous.4open.science/r/OTPR_Supplementary-5F55/README.md).
>
> ## C5. Notation Suggestion
>
> **R5:** Our intention was to use **bold font** to denote variables, while regular font indicates a specific sampled instance of that variable, following [1]. This distinction also helps differentiate between actions at each timestep and those at each denoising step. We acknowledge that this notation might have caused confusion, and we will unify the font style and fix typo errors in the revised manuscript.
>
> [1] Ada, et al. "Diffusion policies for out-of-distribution generalization in offline reinforcement learning." IEEE Robotics and Automation Letters
>
> ## C6. Questions for Authors
>
> **R6:**
>
> **Masking Scheme and Action Discovery**  As we mentioned in our response to the Generalization Concerns (R2), the Mask mechanism is introduced to assist in estimating the OT plan. Nevertheless, it is the compatibility function H(s,a) that truly evaluates the sampled actions and guides the optimization of the diffusion policy. The computation of H is a comprehensive process that includes both the potential functions u and v, along with the Q-values, ensuring robustness. Actions with equally high Q-values are also permitted to achieve correspondingly high H-values.Thus, this does not preclude the model from discovering other good actions.
>
> **Demo-augmented RL Algorithms Performance** Similar reproduction results were also observed in the DPPO paper. RLPD is an online RL algorithm leveraging offline data. Since it does not involve a pre-train process, we set its offline performance to 0. For IBRL, we strictly adhered to the original implementation protocol and its behavioral cloning objective during the offline training stage. Its offline performance is bad, which may be attributed to the presence of noise and multi-modality of data.

---

### Decision · Program_Chairs · 2025-05-01

**Decision:**

Accept (poster)

**Comment:**

This paper develops an optimal transport method for improving diffusion policy optimization. This is an important and timely topic for machine learning. From what I understand, it uses the Q function to define a cost for a transport problem that maps from states to actions, masking to guide matching based on demonstrated state-action pairs, and a resampling strategy to better guide the training process using online environment interactions.  As the reviewers note, the approach is difficult to fully understand, and the authors deserve some responsibility for not more clearly explaining it. There are also numerous typos in the paper (that should mostly be caught by spellcheck), and the authors' rebuttals to reviewer questions often increased rather than decreased reviewer confusion. However, on the positive side, the experimental results often outperform the baselines and the method appears to provide better stability. Since all reviewers are in favor of acceptance, I also recommend acceptance, but hope substantial improvements can be made to the paper's clarity before publication.